# Distinct Bomanins at the Drosophila 55C locus function in resistance and resilience to infections

Yanyan Lou [1,2,3,4,8], Bo Zhang [1,8], Zhiyuan Zhang [1], Yingyi Pan [1], Jianwen Yang[1,2,3], Lu Li[1], Jianqiong Huang[1,2,3], Zihang Yuan [1], Samuel Liegeois [1,2,3], Philippe Bulet [5,6], Rui Xu[1,2,3,7], Li Zi[1] & Dominique Ferrandon [1,2,3]✉

## Abstract

Host defense against many Gram-positive bacteria and fungal pathogens is mainly provided by the Toll-dependent systemic immune response in *Drosophila*. While antimicrobial peptides active against these categories of pathogens contribute only modestly to protection, Bomanin peptides are major effectors of the Toll pathway. Remarkably, flies deleted for the *55C* locus that contains ten *Bomanin* genes are as sensitive as Toll pathway mutant flies to these infections. Yet, the exact functions of single Bomanins in resistance or resilience to infections remain poorly characterized. Here, we have extensively studied the role of these *Bomanin* genes. BomT1 functions in resistance to *Enterococcus faecalis* while playing a role in resilience against *Metarhizium robertsii* infection, like BomS2. BomT1 and BomT2 can prevent the dissemination of *Candida albicans* throughout the host, even though they are not sufficient to confer protection to immunodeficient flies against this pathogen in survival experiments. Furthermore, *BomT1* and *BomBc1* mutants are sensitive to an *Aspergillus fumigatus* ribotoxin. We conclude that *55C* Bomanins have defined albeit sometimes overlapping roles in the different facets of host defense against infections.

**Keywords** Disease Tolerance; Innate Immunity; Host Defense Effectors; Overexpression Transgenes; Genetic Analysis
**Subject Categories** Immunology; Microbiology, Virology & Host Pathogen Interaction; Signal Transduction

## Introduction

The major host defense against many but not all Gram-positive bacteria, pathogenic yeasts, and molds is mediated by the Toll pathway, which regulates one arm of the systemic humoral immune response (Buchon et al, 2014; Ferrandon et al, 2007; Lemaitre and Hoffmann, 2007; Lindsay and Wasserman, 2014). This transmembrane receptor is activated by binding to the Spätzle cytokine, which is itself matured into an active Toll ligand by proteolytic cascades triggered upon sensing microbial cell wall components or the catalytic activity of proteases released by invading pathogenic microorganisms (Liegeois and Ferrandon, 2022). Toll in turn activates a specific NF-κB intracellular signaling cascade that ultimately leads to the induction of expression of likely hundreds of genes, including those encoding antimicrobial peptide (AMP) genes (De Gregorio et al, 2002; Irving et al, 2001; Troha et al, 2018). For instance, Drosomycin and Metchnikowin have been shown to have antifungal activity in vitro (Fehlbaum et al, 1995; Levashina et al, 1995). Yet, even though AMPs play a significant role in the host defense against Gram-negative bacteria, they appear to be rather marginally required for protection against Gram-positive bacteria or fungi (Cohen et al, 2020; Hanson et al, 2019). Remarkably, the deletion of 10 genes of the Bomanin family, members of which had initially been identified by mass-spectrometry analysis of the hemolymph of infected flies (Uttenweiler-Joseph et al, 1998), phenocopied the susceptibility of Toll pathway mutants to these categories of pathogens (Clemmons et al, 2015). The Bomanin family of 12 genes can be divided into three subgroups: short Bomanins (BomSs) that essentially contain a 16-amino-acid conserved Bomanin domain, tailed Bomanins (BomTs) for which the Bomanin domain is prolonged by a 15–82 amino-acid long extension, and bicipital Bomanins (BomBcs) that contain two Bomanin domains separated by a 43–103 amino-acid long linker. Functionally, Bomanins are required for a fungicidal activity against *Nakaseomyces glabrata* (also referred to as *Candida glabrata* (Denning, 2024)) found in the hemolymph of flies and that may require a sufficiently strong expression of BomSs, without much specificity being involved within the distinct BomSs (Lindsay et al, 2018). There is thus some evidence of 55C Bomanins being involved in resistance against *N. glabrata* and actually also against *Enterococcus faecalis* (Clemmons et al, 2015; Lindsay et al, 2018). Of note, BomSs but not BomBcs require the activity of a Toll pathway-regulated protein, Bombardier (Bbd), which may be required for the secretion or stability of BomSs (Lin et al, 2019).

[1]Sino-French Hoffmann Institute, Guangzhou Medical University, Guangzhou, China. [2]Université de Strasbourg, Strasbourg, France. [3]Modèles Insectes de l'Immunité Innée, UPR 9022 du CNRS, Strasbourg, France. [4]Shenzhen Luohu People's Hospital, The Third Affiliated Hospital of Shenzhen University, Shenzhen, China. [5]Université Grenoble Alpes, Institute for Advanced Biosciences, Inserm U1209, CNRS UMR 5309 Grenoble, France. [6]Platform BioPark Archamps, Archamps, France. [7]Department of Pediatrics, The Affiliated Foshan Maternal and Children's Hospital, Guangdong Medical University, Foshan, Guangdong, China. [8]These authors contributed equally: Yanyan Lou, Bo Zhang.
✉E-mail: D.Ferrandon@ibmc-cnrs.unistra.fr

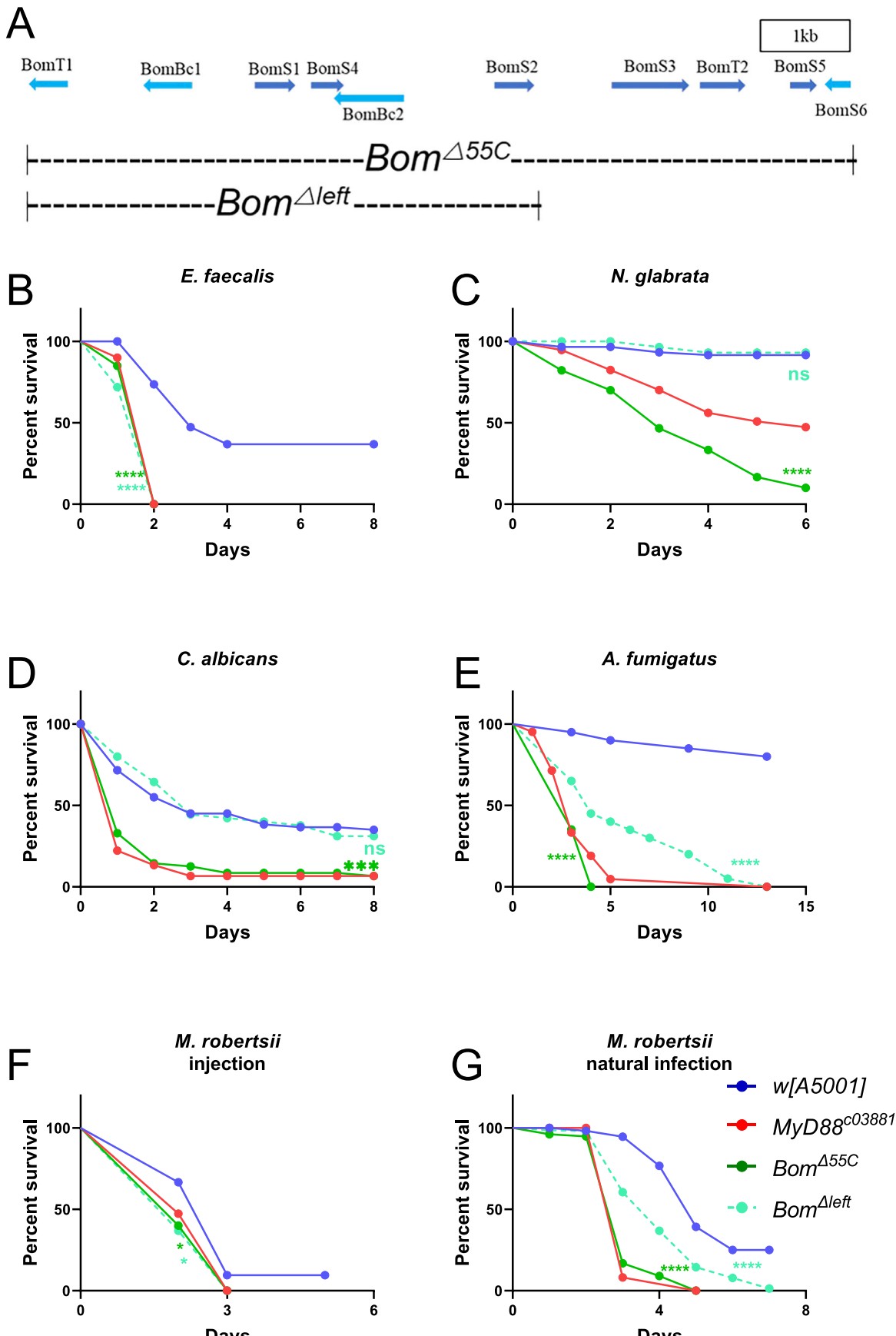

**Figure 1. The 55C *Bomanin* locus is essential for host defense against multiple microbial infections.**

(A) Scheme of the 55C *Bomanin* locus on the second chromosome and map of the *Bom*$^{\Delta55C}$ and *Bom*$^{\Delta left}$ deletions. The 1 kb box gives the scale of the scheme. (B–G) Survival curves of wild-type *w* [A5001], *MyD88*$^{c03881}$, *Bom*$^{\Delta left}$, and *Bom*$^{\Delta55C}$ flies after *E. faecalis* (B), *N. glabrata* (C), *C. albicans* (D), *A. fumigatus* (E), *M. robertsii* (F) injection, and *M. robertsii* natural infection (G). Data information: In (B–G), three experiments were performed at different times, and each experiment used biological triplicates of 20 flies in parallel. Pooled data were analyzed between infected-mutant and - *w* [A5001] fly using the Log-Rank test; ns, no significant difference. (B, E, G) *Bom*$^{\Delta left}$ and *Bom*$^{\Delta55C}$ flies: *P* < 0.0001; (C) *Bom*$^{\Delta55C}$ flies: *P* < 0.0001; (D) *Bom*$^{\Delta55C}$ flies, *P* = 0.0008; (F) *Bom*$^{\Delta55C}$ flies, *P* = 0.048, *Bom*$^{\Delta left}$ flies, *P* = 0.035. Source data are available online for this figure.

We have recently reported that an important component of host defense against the opportunistic fungal pathogen *Aspergillus fumigatus* is the protection against secreted mycotoxins, which is partially mediated by specific Bomanins (Xu et al, 2023), in keeping with another study extending the concept to the defense against secreted virulence factors by another family of Toll pathway effectors, BaramicinA-derived peptides (Huang et al, 2023). Thus, 55C Bomanins are also involved in resilience (also referred to as disease tolerance) to infections, as in this case, they are not directly targeting the microorganisms as AMPs do (Duneau et al, 2017; Ferrandon, 2013; Soares et al, 2017).

While the deletion of subsets of *Bom* genes revealed important information on the function of this family of effector peptides (Clemmons et al, 2015), little is known with respect to the function of individual *Bom* genes (Chapman et al, 2020; Smith et al, 2023; Xu et al, 2023). Here, we use complementary genetic strategies to implement the dissection of *55C Bomanin* function in the host defense against a selected set of microbes representing distinct categories of pathogens.

# Results

## The 55C Bomanin locus is required in the host defense against several microbial infections

We first checked that we could reproduce the previously published data (Clemmons et al, 2015) when isogenizing the 55C deletions, *Bom*$^{\Delta55C}$ and *Bom*$^{\Delta left}$, in our wild-type genetic background *w* [A5001] (Thibault et al, 2004) (Fig. 1A). As reported previously, we found that *Bom*$^{\Delta55C}$ mutant flies were highly susceptible to challenges with *E. faecalis* and *N. glabrata*. *Bom*$^{\Delta left}$ mutant flies were either sensitive (*E. faecalis*) or behaved as wild-type flies (*N. glabrata*) as expected (Fig. 1B,C). We next tested additional pathogens. The two *Bom* deficiencies exhibited phenotypes after the injection of *C. albicans* that were similar to those observed after *N. glabrata* challenge, even though the two pathogenic yeasts are evolutionarily distant, the former one being dimorphic (Fig. 1D). Whereas the *Bom*$^{\Delta55C}$ line was as sensitive as the Toll pathway mutant line *MyD88*, *Bom*$^{\Delta left}$ displayed an intermediate sensitivity phenotype to an *A. fumigatus* challenge (Fig. 1E). In keeping with results reported for the sensitivity of *Bom*$^{\Delta55C}$ to the entomopathogenic fungus *Beauveria bassiana* (Hanson et al, 2019), both *Bom*-deficiency lines succumbed rapidly at the same rate to injected spores of a related fungus, *Metarhizium robertsii* (Fig. 1F). In contrast, upon a natural infection (see below) with the same fungus, *Bom*$^{\Delta left}$ mutants exhibited an intermediate susceptibility phenotype (Fig. 1G).

*Bom*$^{\Delta55C}$ mutant flies were remarkably at least as sensitive as *MyD88* flies to these infections, in keeping with previous studies (Clemmons et al, 2015; Hanson et al, 2019; Smith et al, 2023).

As the host defense encompasses both humoral and cellular immunity, we tested whether *Bom*$^{\Delta55C}$ mutant flies were defective for the phagocytosis of live *N. glabrata* (Liegeois et al, 2020). We did not observe any significant difference in the uptake of these live microorganisms (Fig. EV1).

## Genetic strategies to dissect the different functions of the Bomanins encoded at the 55C locus

The *Bom* genes located at the *55C* locus are required for both resistance and resilience to infections (Clemmons et al, 2015). It is currently not known whether specific *Bomanin* genes are specific to host defense against a given pathogen, whether some are particularly involved in resistance or resilience, whether they may play redundant roles, or are required for a "cocktail" effect. Here, we have implemented a dual genetic strategy to address these issues.

First, we have attempted to study the loss-of-function sensitivity phenotype of single *Bom* genes at 55C after a challenge with either the Gram-positive bacterium *E. faecalis*, the pathogenic yeasts *N. glabrata* and *C. albicans*, the filamentous fungus *A. fumigatus*, or the entomopathogenic fungus *M. robertsii* in two infection models, injection and natural infection. No macro-injury is involved in *M. robertsii* natural infection as the conidia differentiate an appressorium that allow them to pierce through the cuticle and to penetrate the body cavity. The fungus exhibits distinct properties according to the infection route, both in terms of morphology (hyphal bodies in the injection *vs.* filaments in the natural infection) and relevant host defenses, the Toll pathway being important in both models (Wang, 2020). We have generated deletions, knock-out (KO) or knock-in (KI) mutants using CRISPR-Cas9 technology for *BomT1*, *BomBc1*, *BomS2*, *BomT2*, and *BomS5* (Appendix Figs. S1–S6). We expect to have generated null mutants for *BomT1*, *BomS2* (two independent mutants), *BomT2* (*mCherry* KI, one deletion plus one indel line), and *BomS5*. Of note, the deletion of *BomS5* severely affects the expression of *BomT2* while increasing that of *BomS6* (Appendix Fig. S6C). Likewise, the *BomT2-mCherry* KI mutation leads to a lower level of induction by *Micrococcus luteus* for six of the ten 55C *Bom* genes, plus *Drosomycin* (Appendix Fig. S4C), whereas the expression of seven 55C genes but not of *Drosomycin* is decreased in the *BomT2*$^{\Delta tail}$ indel line (Appendix Fig. S5C). An indel mutation has been generated after the region encoding the first Bomanin domain for *BomBc1*; this mutant may thus still be able to express a truncated protein that would belong to the BomT category. In addition, we have also tested knock-down lines for *BomT1*, *BomBc1*, *BomS4*, and *BomBc2* by RNA interference (Appendix Fig. S7). We have failed to obtain mutant lines affecting the expression of *BomS1*, *BomS3*, and *BomS6*.

Second, we have initiated overexpression strategies to bypass any redundancy issues by overexpressing each gene from

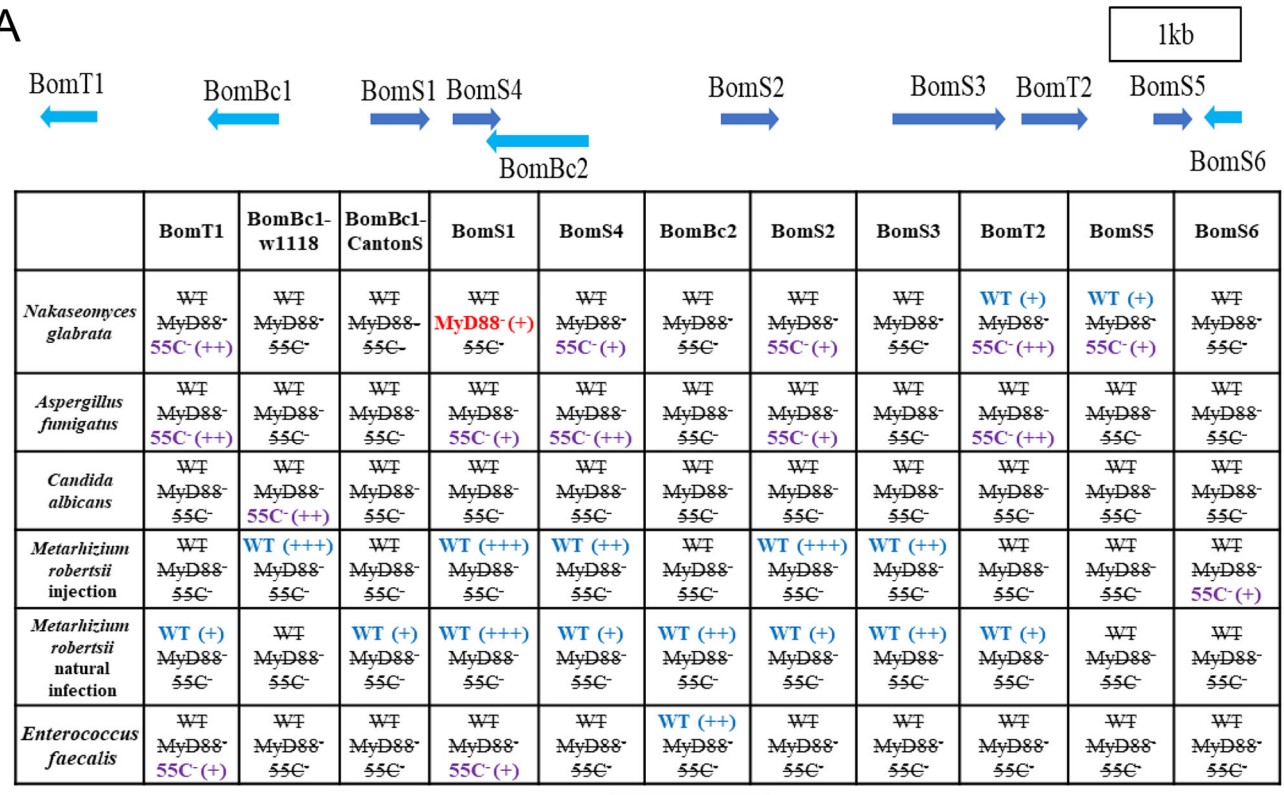

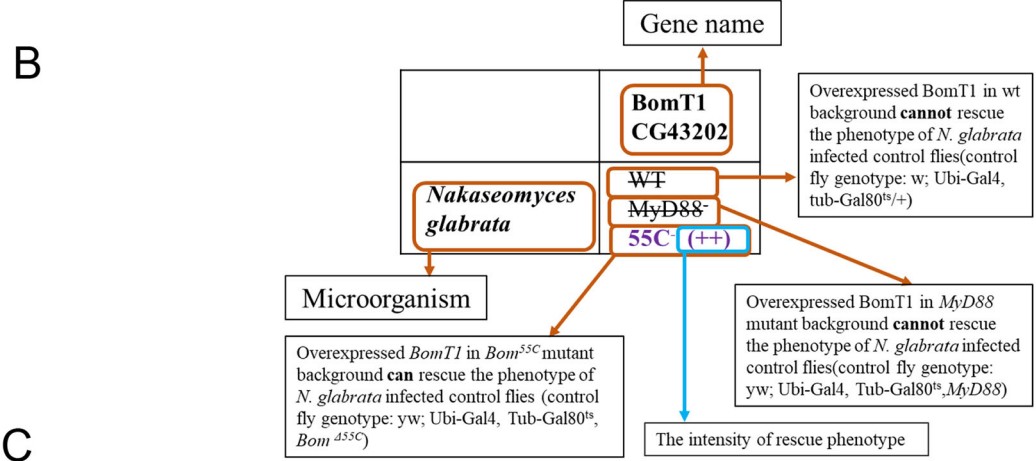

**Figure 2.  Summary of survival experiments for loss-of-function or overexpression in wild-type, MyD88, or Bom$^{\Delta55C}$ backgrounds of single 55C Bomanin genes.**

(A) This panel recapitulates the results of survival experiments with multiple pathogens obtained by the overexpression approach. The qualitative level of rescue is indicated by the (+) signs as assessed from multiple survival experiments, as documented in Figs. 3C, 4E, EV2B–F, EV3, Appendix Fig. S10A,G and S11A. When no rescue or effect was observed, the ~~WT~~ (wild-type background), ~~MyD88~~ (MyD88 background) or ~~55C~~ (Bom$^{\Delta55C}$ background) notation was adopted. The text in red 'MyD88(+)' for BomS1 means that BomS1 overexpression in MyD88 mutant background flies not only did not rescue the N. glabrata susceptibility phenotype but made it actually more susceptible to this infection, as compared to MyD88$^{c03881}$ flies. (B) Key to the annotation of each composite cell of (A). (C) This panel recapitulates the results of survival experiments with multiple pathogens on loss-of-function "mutants" of 55C Bomanin genes. The qualitative level of sensitivity to a given infection is indicated by the (+) signs as assessed from multiple survival experiments, as documented in Figs. 3A,B,F, 4A, EV2A, and Appendix Fig. S10B–F. The degree of confidence in the phenotype depending on whether we obtained similar results with independent lines, e.g., same sensitivity in KO and KD mutants (BomT1), is indicated by the intensity of the green background from strong (dark) to weak (light). NS no significant difference, KI knock-in mutant, KD knock-down obtained by RNAi silencing with the indicated line, KO knock-out. Source data are available online for this figure.

transgenes constructed for expression under the control of *UAS* enhancer sequences (Brand and Perrimon, 1993). *Bom* genes were overexpressed first in a wild-type (Appendix Fig. S8) and second in a *MyD88* immunodeficient background. However, upon checking the expression of the BomSs in the hemolymph by MALDI-TOF mass-spectrometry (MS) (Uttenweiler-Joseph et al, 1998), we failed to observe the expected peaks in *MyD88* flies (we had to use *MyD88* flies to check that the overexpression constructs were working as expected as the signal from the overexpression strategy is masked in wild-type flies by that from the endogenous peptides). In addition, no ectopic expression was detected in the absence of an immune challenge in wild-type flies overexpressing *BomS* genes, even though they are induced at the transcriptional level, albeit less than after *M. luteus* septic injury (Appendix Fig. S8), suggesting that an immune challenge may be necessary for the translation or secretion of AMPs into the hemolymph. We reasoned that the lack of detection of BomS peptides from overexpressed *BomS* in *MyD88* mutant flies might result from the absence of induction of other *MyD88*-dependent genes that might be required for their translation, secretion or stability in the hemolymph. Indeed, the role of Bombardier in the secretion or stability of BomS peptide was reported while this work was being pursued (Lin et al, 2019). Importantly, *Bbd* is a Toll-regulated gene. We therefore decided to test each *Bom* gene by its overexpression for a rescuing activity of the sensitivity of *Bom$^{\Delta55C}$* deficiency flies to various microbial challenges, that is by testing the effects of *Bom* gene overexpression in a *Bom$^{\Delta55C}$* background. As expected, we then detected the expression of several BomS peptides by MALDI-TOF MS in *Bom$^{\Delta55C}$*-deficiency flies (Appendix Fig. S9A). The overexpression of the other *Bom* genes in the *Bom$^{\Delta55C}$* deficiency was checked at the transcript level (Appendix Fig. S9B).

The results we have obtained are summarized in Fig. 2A (overexpression of single 55C *Bom* genes in wild-type, *MyD88$^{c03881}$*, and *Bom$^{\Delta55C}$* flies; a key to understanding Fig. 2A is provided in Fig. 2B), and C (loss-of-function approach). In the following, we shall describe in more detail the most outstanding results we have obtained.

### *BomT1* is required but not sufficient for resistance against *E. faecalis*

When challenged with *E. faecalis*, the *BomT1-Gal4* knock-in (KI) null mutant flies succumbed at a rate that was almost as fast as that of *MyD88* mutants (Fig. 3A). A similar, albeit milder, phenotype was observed when *BomT1* was silenced ubiquitously only at the adult stage (Fig. 3B). Of note, the overexpression of *BomT1* in wild-

type flies did not protect them from *E. faecalis* (Fig. 2A; Appendix Fig. S10A); however, it conferred a partial protection to *Bom$^{\Delta55C}$* deletion mutants, suggesting it may need to act in concert with another 55C *Bomanin* gene for protection to wild-type levels (Fig. 3C). We have so far failed to identify such a gene in our loss-of-function approach, even though one of the two null *BomS2* mutant line displayed a sensitivity to *E. faecalis* (Figs. 2A and EV2A). We also noted that *BomS1* overexpression also partially rescued the *Bom$^{\Delta55C}$* deletion sensitivity phenotype (Figs. 2A and EV2B). Unexpectedly, *BomBc2* overexpression provided a degree of protection against *E. faecalis* in wild-type but not *Bom$^{\Delta55C}$* flies (Figs. 2A and EV2C), suggesting it may also act in concert with another 55C *Bom* gene, *BomT1* being the best candidate.

We next investigated whether the *E. faecalis* burden was altered in *BomT1-Gal4* KI single flies. As shown in Fig. 3D, the bacterial load was higher than in wild-type flies from 6 h onwards, implying that *BomT1* is required for resistance. We also measured the number of bacteria present in single flies within 30 min of their death and found that both *MyD88* and *BomT1* displayed an increased titer in the *w*[A5001] background (Fig. 3E), suggesting that more bacteria are needed to kill the *BomT1* and *MyD88* immunodeficient flies. It is not clear, however, whether this finding reflects an increased resilience to *E. faecalis* infection.

### *BomT1* may be involved in resilience to *M. robertsii* natural infection

We found that *BomT1-Gal4* KI mutants were significantly more susceptible to *M. robertsii* natural infection, but not to the injection of its spores (Figs. 2C and 3F; Appendix Fig. S10B). Of note, in contrast to the *E. faecalis* sensitivity phenotype, *BomT1*-silenced flies did not display any increased susceptibility to *M. robertsii* natural infection. This may reflect a hypomorphic effect of the RNAi approach, even though the silencing appeared rather strong at the transcript level (Appendix Fig. S7A). *M. robertsii* would also need to be more sensitive than *E. faecalis* to any remaining BomT1 in these silenced lines. In contrast to *MyD88* flies, the fungal load in *BomT1-Gal4* KI single flies did not appear to vary at any time point (Fig. 3G). In terms of fungal burden within 30 min of death, there was no measured difference between wild-type and immunodeficient flies (Fig. 3H). Thus, our data are compatible with the hypothesis that BomT1 functions in resilience against *M. robertsii* natural infection.

Finally, *BomT1* overexpression in wild-type but not in *Bom$^{\Delta55C}$* flies provided a moderate degree of protection against this

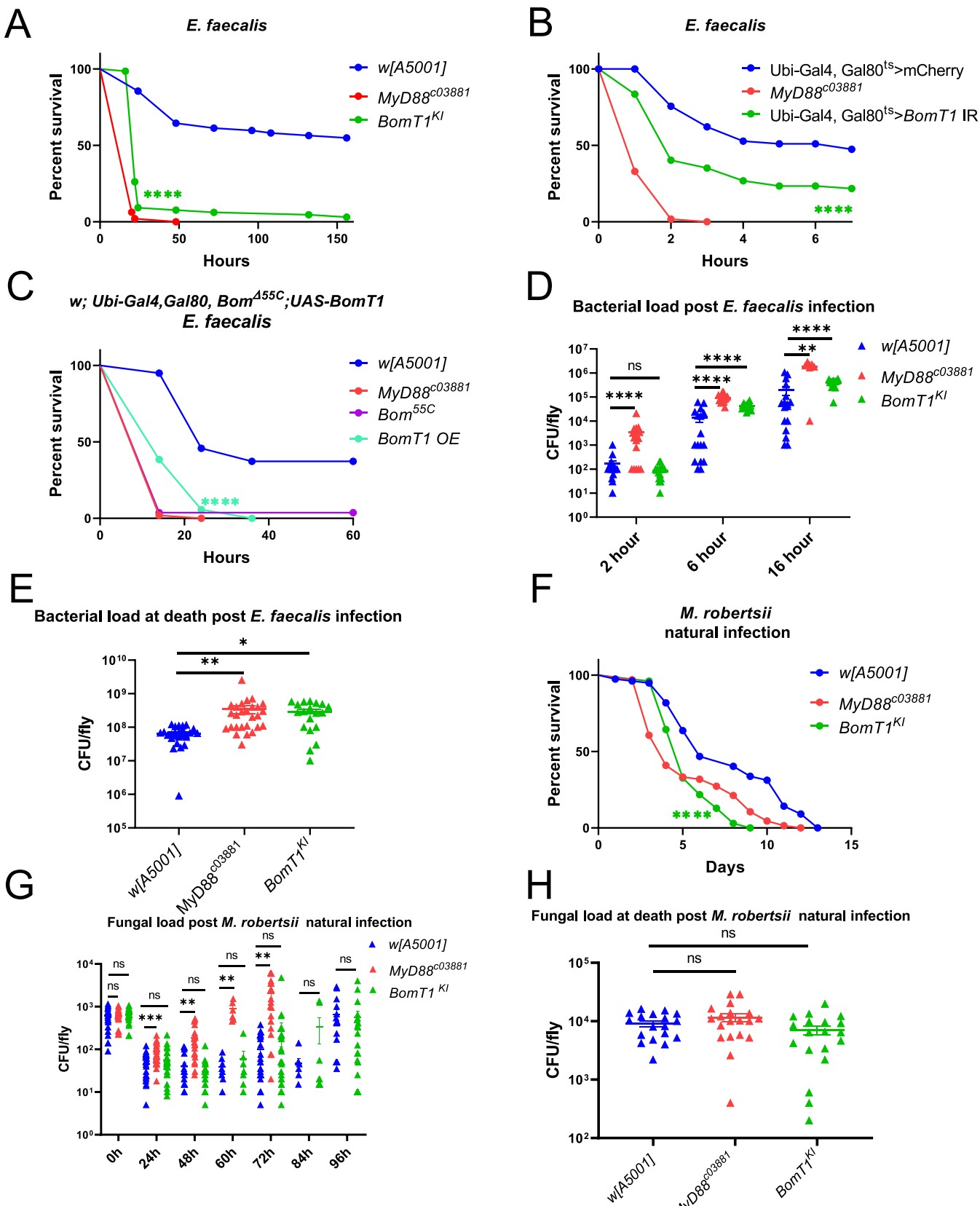

**Figure 3. BomT1 partially mediates resistance to *E. faecalis* and resilience to *M. robertsii* natural infection.**

(A, B) Survival curves of *BomT1-Gal4* KI (*BomT1^KI^*) mutant flies (A) or flies with ubiquitous silencing of *BomT1* at the adult stage (B) after *E. faecalis* (OD$_{600}$ = 0.1, 4.6 nL/fly) injection. (C) survival curves of flies ubiquitously overexpressing *BomT1* (OE) at the adult stage in *Bom^Δ55C^* deletion background challenged by *E. faecalis* injection. (D) Bacterial load of *BomT1^KI^* mutant, w [A5001] and *MyD88^c03881^* single flies 2, 6, and 16 h post *E. faecalis* injection. (E) Bacterial load within 30 min of death of *BomT1^KI^* mutant, w [A5001] and *MyD88^c03881^* flies post *E. faecalis* injection. (F) Survival curves of *BomT1^KI^*, *MyD88^c03881^* mutant and w [A5001] flies after *M. robertsii* (10⁵ spores/mL, 5 mL/ group) natural infection. (G) Fungal load of *BomT1^KI^* mutant, w [A5001] and *MyD88^c03881^* flies at different time points post *M. robertsii* natural infection. (H) Fungal load within 30 min of death of *BomT1^KI^* single mutant flies, w [A5001] and *MyD88^c03881^* flies post *M. robertsii* natural infection. Data information: three experiments were performed at different times; each survival experiment used biological triplicates of 20 flies in parallel. (A, B, C, F) The pooled data were analyzed between infected-mutant and - w [A5001] fly using the Log-Rank test. (D, E, G, H) The data are presented as means ± SEM and analyzed using the ANOVA (one-way) with Tukey's multiple comparisons post hoc test. (A) *BomT1^KI^* flies, P < 0.0001. (B) Ubiquitous silencing of *BomT1* at the adult stage flies, P < 0.0001; (C) Overexpressing *BomT1* (OE) at the adult stage in *Bom^Δ55C^* deletion background flies, P < 0.0001. (D) *MyD88^c03881^* flies at 2 h, 6 h, and 16 h: P < 0.0001, P < 0.0001, P = 0.0031; *BomT1^KI^* flies at 2 h: P = 0.99, at 6 h: P < 0.0001, at 16 h: P < 0.0001. (E) *MyD88^c03881^* flies: P = 0.0035; *BomT1^KI^* flies: P = 0.031. (F) *BomT1^KI^* mutant flies: P < 0.0001. (G) *MyD88^c03881^* flies at 24 h: P = 0.0003, at 48 h: P = 0.0032, at 60 h: P = 0.0034, at 72 h: P = 0.0006; *BomT1^KI^* flies at 24 h: P = 0.62, at 48 h: P = 0.87, at 60 h: P = 0.98, at 72 h: P = 0.83, at 84 h: P = 0.86, at 96 h: P = 0.32. (H) *MyD88^c03881^* flies: P = 0.36, *BomT1^KI^* flies: P = 0.45. Source data are available online for this figure.

entomopathogenic fungus, suggesting it may act in concert with another *55C Bom* (Fig. 2A). In this respect, we discovered that the overexpression of *BomBc1* (Canton-S isoform), *BomS4*, *BomS2*, or *BomT2* protected flies from *M. robertsii* natural infection to the same extent as *BomT1* overexpression in wild-type flies (Fig. 2A). Furthermore, *BomBc2*, *BomS3*, and especially *BomS1* overexpression provided an even stronger level of protection (Figs. 2A and EV2D–F).

## BomS2 may be involved in resilience against injected *M. robertsii*

In opposition to *BomT1*, we found that the two *BomS2* null mutants were sensitive to injected *M. robertsii* spores but not to a natural infection with the same pathogen (Figs. 2C and 4A). While we measured a somewhat increased fungal load at three days in *BomS2^ΔKO6^* mutant flies, we failed to reproduce this observation in *BomS2^ΔKO36^* flies as well as in *BomS2^ΔKO6^*/*BomS2^ΔKO36^* trans-heterozygous flies (Fig. 4B–D). Thus, *BomS2* may be involved in resilience against this infectious challenge.

The overexpression of *BomS2* in wild-type but not in *Bom^Δ55C^* flies protected them from the effects of *M. robertsii* injection at two inoculum doses (Figs. 2A and 4E). Actually, we noted a similar level of protection conferred by the overexpression, also in adult wild-type flies, of *BomS1* or *BomBc1* whereas that of *BomS3* or *BomS4* offered a more limited level of defense (Fig. EV3). Of note, *BomBc1* expresses distinct polymorphism isoforms in $w^{1118}$ and in *Canton-S* wild-type background (Appendix Fig. S2D). While the overexpression of the former enhanced the survival solely against injected *M. robertsii*, that of the latter did so only upon natural infection, an indication of the specificity of the roles of these two isoforms depending on the infection route of this pathogen (Fig. 2A).

Unexpectedly, we detected that *BomS2^ΔKO36^* but not *BomS2^ΔKO6^* mutant flies displayed an increased sensitivity to *E. faecalis* (Figs. 2C and EV2A), possibly in keeping with the report that *BomS2* silencing led to sensitivity to another, mild, Gram-positive bacterial pathogen, *Lysinibacillus fusiformis* (Smith et al, 2023).

## Bomanin-mediated host defenses against *A. fumigatus*, *N. glabrata*, or *C. albicans*?

We have not identified any *Bom* gene with a clear-cut sensitivity phenotype to any of these three pathogens using the set of mutants/

RNAi lines we have tested. The sensitivity of one *BomBc1* RNAi line to *C. albicans* was not confirmed in an independent line that appears as efficient in terms of silencing at the transcript level (Fig. 2C, Appendix Figs. S7B and S10C,D). Similarly, the susceptibility to a *N. glabrata* challenge of the *BomT2* indel line, which may still express a protein lacking the BomT2 tail, was not observed with the two null mutant lines (Fig. 2C). The *BomT2^indel^* phenotype might be accounted for by the decreased inducibility of a majority of *55C Bom* genes observed in this mutant (Appendix Fig. S5).

We note that with respect to *N. glabrata* and *A. fumigatus* infections, the *Bom^Δ55C^* sensitivity phenotype can be rescued in general to a rather mild degree by the overexpression of a similar set of *Bom* genes, that is both *BomT* genes and *BomS* genes, with *BomS1* overexpression being able to protect the *Bom^Δ55C^*-deficiency flies only against *A. fumigatus* whereas *BomS5* overexpression provided some defense to these flies only against *N. glabrata* (Fig. 2A).

Unexpectedly, the overexpression of *BomS1* in *MyD88* flies made them more sensitive to a *N. glabrata* challenge (Appendix Fig. S11A). It has been previously reported that Bombardier is involved in resilience by protecting the flies from the noxious effects of *BomS* genes when not secreted/stabilized in the hemolymph (Lin et al, 2019). Thus, BomS1 may contribute to the *Bbd* resilience phenotype. We have noticed in this respect that the continuous overexpression throughout the life cycle of the fly of *BomS1* led to a developmental phenotype characterized by Stubble-like bristles (Appendix Fig. S11B–D). Thus, the constitutive expression of *BomS1* does interfere to some extent with development, and possibly elsewhere in a less visible manner. Note, however, that thanks to the Gal4-Gal80^ts^ system (McGuire et al, 2004), we overexpress *Bom* genes only at the adult stage in our overexpression approach (Fig. 2A), thereby bypassing any interference with normal development.

In contrast to the two infections above, the overexpression of only one *Bom* gene, the $w^{1118}$ *BomBc1* isoform, protected to a mild degree *Bom^Δ55C^* flies against *C. albicans* infection (Fig. 2A, Appendix Fig. S10G).

## A role for Bomanins in preventing the dissemination of *C. albicans* throughout the fly body

Survival assays may not reflect the full palette of Bomanin functions. We reasoned that one aspect of infection is the

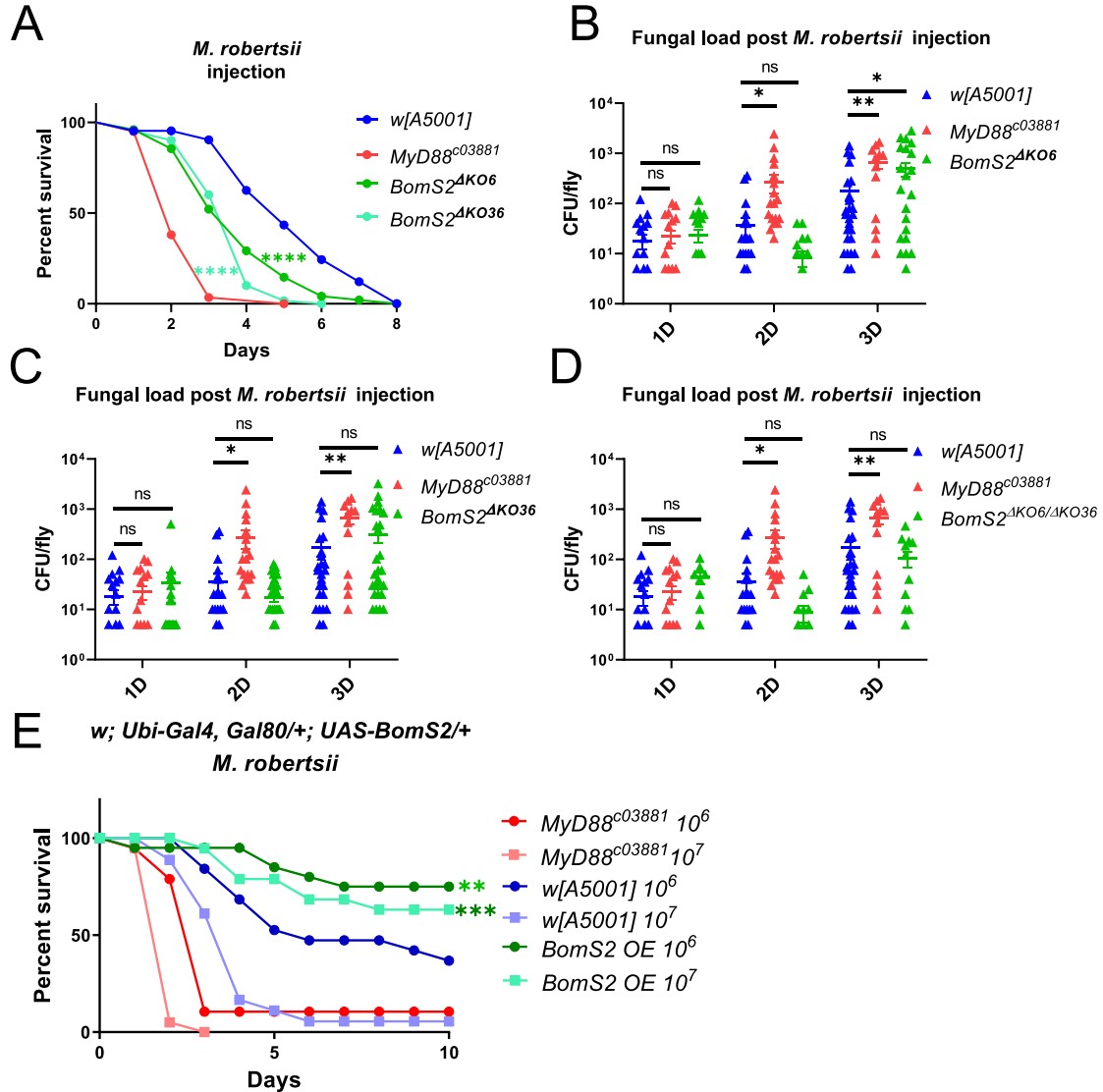

**Figure 4. BomS2 partially mediates resilience to injected _M. robertsii_ spores.**

(A) Survival curves of two _BomS2_ null mutants, _BomS2ΔKO6_ and _BomS2ΔKO36_ after _M. robertsii_ ($10^7$ spores/mL, 4.6 nL/fly) injection. (B–D) Fungal load of _BomS2ΔKO_ (B), _BomS2ΔKO36_ (C), and transheterozygous _BomS2ΔKO6/BomS2ΔKO36_ mutant flies (D) post _M. robertsii_ injection at different time points. (E) Survival experiments of _BomS2_-overexpressing (OE) flies in a wild-type background to _M. robertsii_ injection. Data information: more than three experiments were performed at different times; each survival experiment used biological triplicates of 20 flies in parallel. (A, E) The pooled data were analyzed between infected-mutant and - _w_ [A5001] fly using the Log-Rank test. (B–D) One-way ANOVA with Tukey's post hoc test. (A) _BomS2ΔKO6_ and _BomS2ΔKO36_ flies: $P < 0.0001$; (B) _MyD88c03881_ flies at 1st day: $P = 0.85$, at 2nd day: $P = 0.041$, at 3rd day: $P = 0.0089$; _BomS2ΔKO6_ flies at 1st day: $P = 0.99$, at 2nd day: $P = 0.081$, at 3rd day: $P = 0.049$; (C) _MyD88c03881_ flies at 1st day: $P = 0.85$, at 2nd day: $P = 0.041$, at 3rd day: $P = 0.0089$; _BomS2ΔKO36_ flies at 1st day: $P = 0.75$, at 2nd day: $P = 0.068$, at 3rd day: $P = 0.090$; (D) _MyD88c03881_ flies at 1st day: $P = 0.85$, at 2nd day: $P = 0.041$, at 3rd day: $P = 0.0089$; Transheterozygous _BomS2ΔKO6 /BomS2ΔKO36_ mutant flies at 1st day: $P = 0.17$, at 2nd day: $P = 0.49$, at 3rd day: $P = 0.11$; (E) _BomS2_-overexpressing (OE) flies in $10^6$: $P = 0.0039$, in $10^7$: $P = 0.0004$. Source data are available online for this figure.

dissemination of the pathogen away from its initial infection site. As a bright GFP-expressing strain of _C. albicans_ was available, we examined how _C. albicans_ behaved within the fly once injected in the thorax. The pathogenic yeast remained mostly at the injection site in _w_ [A5001], yet they managed to form hyphae. In striking contrast, we observed a dissemination of _C. albicans_ throughout the body of _BomΔ55C_ flies, with hyphae detected in all three tagmata (Fig. 5A).

We next complemented the _BomΔ55C_ phenotype by overexpressing each 55C _Bom_ gene one-by-one. Both _BomT1_ and _BomT2_, and to a lesser extent _BomBc1_ prevented significantly the dissemination of _C. albicans_ throughout the body of _BomΔ55C_ flies. Conversely, there was a trend for the _BomBc1_ indel mutant but not for the _BomT1-Gal4_ KI mutant flies to allow some dissemination of _C. albicans_ to the abdomen in an otherwise wild-type background (Fig. 5B).

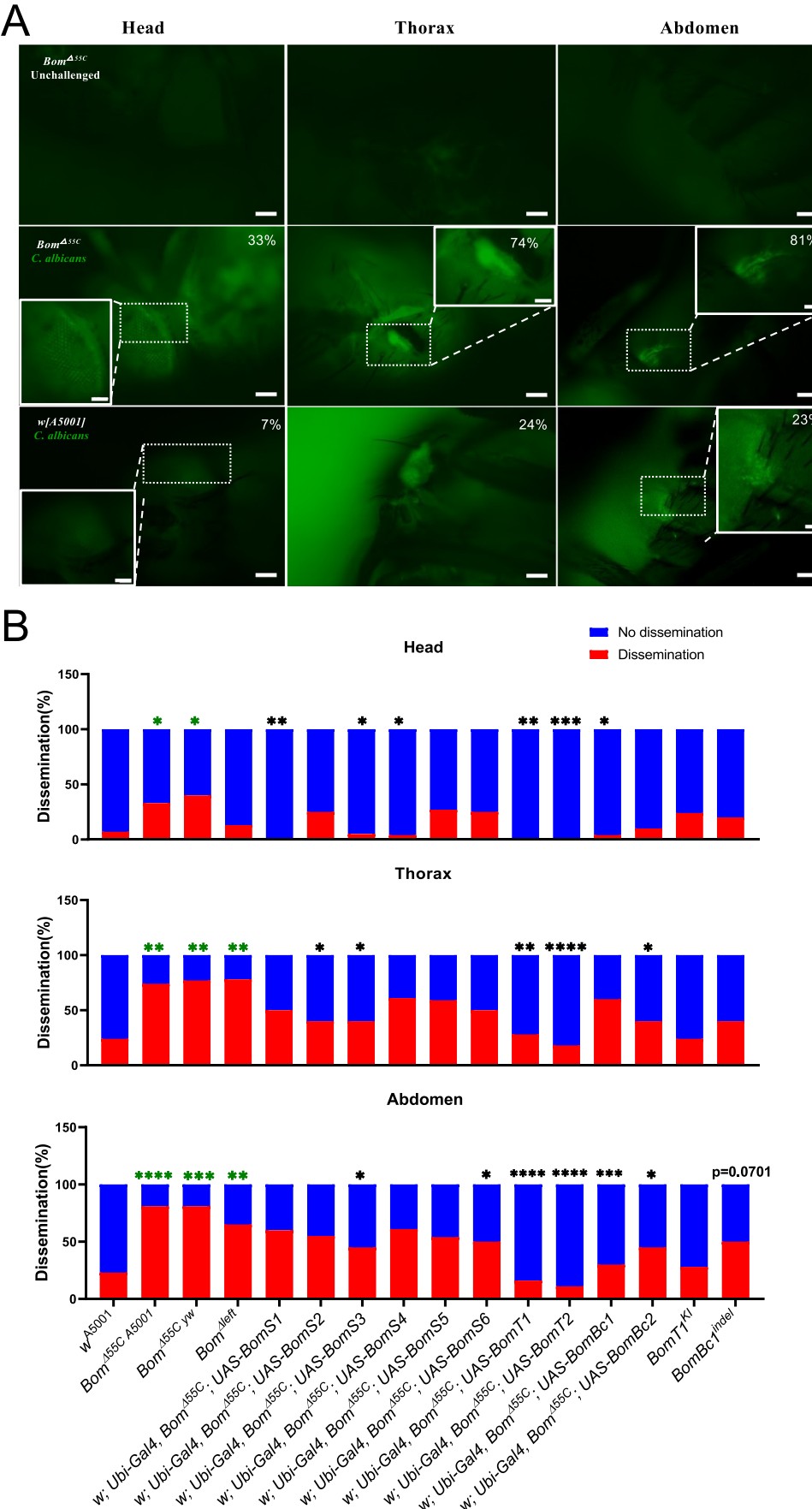

**Figure 5.** *55C Bomanins* **prevent the dissemination of** *C. albicans* **throughout the fly body.**

(A) Detection of GFP-expressing *C. albicans* in *Bom^Δ55C* (middle row) and *w*[A5001] (bottom row) tagmata. Uninjected *Bom^Δ55C* (top row) serve as negative controls. The apparent background signal is coming from *C. albicans* infection foci outside of the focal plan. Insets show blow-ups of the indicated areas (dashed boxes). Scale bars are 0.2 mm and 50 μm for insets. % numbers indicate the fraction of analyzed flies that displayed a detectable signal (B). (B) Pooled data of *C. albicans* conidia formation in different flies at several optical fly sections. *w* [A5001]: wild-type flies. *Bom^Δ55C*(A5001): *Bom^Δ55C* backcrossed to a *w* [A5001] background. *Bom^Δ55C*(yw): *Bom^Δ55C* in its initial *yw* background. *BomS1, BomS2, BomS3, BomS4, BomS5, BomS6, BomT1, BomT2, BomBc1, BomBc2*: adult flies ubiquitously overexpressing *BomS1, BomS2, BomS3, BomS4, BomS5, BomS6, BomT1, BomT2, BomBc1* (*w^1118* isoform), *BomBc2* in a *Bom^Δ55C* background. *BomT1-Gal4* KI (*BomT1^KI*): *BomT1* mutant. *BomBc1^indel*: hypomorphic mutant. Data information: each group contained at least 20 flies. B: The data were analyzed using Fisher's exact test. Black *: comparison to *Bom^Δ55C*; Green *: comparison to *w* [A5001]. Head: *Bom^Δ55C*(A5001), $P = 0.039$, *Bom^Δ55C*(yw), $P = 0.013$, *Bom^Δleft*, $P = 0.66$, w; Ubi-Gal4, *Bom^Δ55C*; UAS-BomS1, $P = 0.0061$, w; Ubi-Gal4, *Bom^Δ55C*; UAS-BomS2, $P = 0.75$, w; Ubi-Gal4, *Bom^Δ55C*; UAS-BomS3, $P = 0.029$, w; Ubi-Gal4, *Bom^Δ55C*; UAS-BomS4, $P = 0.014$, w; Ubi-Gal4, *Bom^Δ55C*; UAS-BomS5, $P = 0.76$, w; Ubi-Gal4, *Bom^Δ55C*; UAS-BomS6, $P = 0.75$, w; Ubi-Gal4, *Bom^Δ55C*; UAS-BomT1, $P = 0.0018$, w; Ubi-Gal4, *Bom^Δ55C*; UAS-BomT2, $P = 0.0007$, w; Ubi-Gal4, *Bom^Δ55C*; UAS-BomBc1, $P = 0.014$, w; Ubi-Gal4, *Bom^Δ55C*; UAS-BomBc2, $P = 0.086$, *BomT1^KI*, $P > 0.99$, *BomBc1^indel*, $P = 0.38$. Throax: *Bom^Δ55C*(A5001), $P = 0.0058$, *Bom^Δ55C*(yw), $P = 0.0042$, *Bom^Δleft*, $P = 0.0037$, w; Ubi-Gal4, *Bom^Δ55C*; UAS-BomS1, $P = 0.13$, w; Ubi-Gal4, *Bom^Δ55C*; UAS-BomS2, $P = 0.034$, w; Ubi-Gal4, *Bom^Δ55C*; UAS-BomS3, $P = 0.034$, w; Ubi-Gal4, *Bom^Δ55C*; UAS-BomS4, $P = 0.37$, w; Ubi-Gal4, *Bom^Δ55C*; UAS-BomS5, $P = 0.36$, w; Ubi-Gal4, *Bom^Δ55C*; UAS-BomS6, $P = 0.13$, w; Ubi-Gal4, *Bom^Δ55C*; UAS-BomT1, $P = 0.0019$, w; Ubi-Gal4, *Bom^Δ55C*; UAS-BomT2, $P < 0.0001$, w; Ubi-Gal4, *Bom^Δ55C*; UAS-BomBc1, $P = 0.37$, w; Ubi-Gal4, *Bom^Δ55C*; UAS-BomBc2, $P = 0.034$, *BomT1^KI*, $P = 0.54$, *BomBc1^indel*, $P = 0.76$. Abdomen: *Bom^Δ55C*(A5001), $P < 0.0001$, *Bom^Δ55C*(yw), $P = 0.0001$, *Bom^Δleft*, $P = 0.0041$, w; Ubi-Gal4, *Bom^Δ55C*; UAS-BomS1, $P = 0.19$, w; Ubi-Gal4, *Bom^Δ55C*; UAS-BomS2, $P = 0.06$, w; Ubi-Gal4, *Bom^Δ55C*; UAS-BomS3, $P = 0.013$, w; Ubi-Gal4, *Bom^Δ55C*; UAS-BomS4, $P = 0.13$, w; Ubi-Gal4, *Bom^Δ55C*; UAS-BomS5, $P = 0.0625$, w; Ubi-Gal4, *Bom^Δ55C*; UAS-BomS6, $P = 0.030$, w; Ubi-Gal4, *Bom^Δ55C*; UAS-BomT1, $P < 0.0001$, w; Ubi-Gal4, *Bom^Δ55C*; UAS-BomT2, $P < 0.0001$, w; Ubi-Gal4, *Bom^Δ55C*; UAS-BomBc1, $P = 0.0005$, w; Ubi-Gal4, *Bom^Δ55C*; UAS-BomBc2, $P = 0.013$, *BomT1^KI*, $P = 0.75$, *BomBc1^indel*, $P = 0.07$. Source data are available online for this figure.

## Implication of the 55C Bomanin cluster in the host defense against *A. fumigatus* mycotoxins

We have previously reported that *Bom^Δ55C* are sensitive to the injection of two mycotoxins secreted by *A. fumigatus*, namely the ribotoxin protein restrictocin and the family of fumitremorgin/verruculogen compounds. We had also reported that *A. fumigatus* mutants lacking the locus encoding restrictocin or lacking the fumitremorgin biosynthetic cluster were only mildly less virulent than the wild-type fungus in *MyD88*-immunodeficient flies (Xu et al, 2023). We have extended these observations to the *Bom^Δ55C* and *Bom^Δleft* fly lines. Whereas we still observed a reduced virulence in the *Bom^Δ55C* line, as for *MyD88* flies, we no longer measured a significant difference in the virulence of the mycotoxin-defective fungi as compared to wild-type fungi in the *Bom^Δleft* line (Fig. EV4). This suggests that the four genes remaining in *Bom^Δleft* flies, namely *BomS3, BomT2, BomS5*, and *BomS6* are sufficient to protect the flies against restrictocin and verruculogen/fumitremorgins.

We also tested directly the difference in sensitivity between *Bom^Δ55C* and *Bom^Δleft* after exposure to restrictocin, fumitremorgin B or verruculogen. Whereas verruculogen killed equally well both deletion lines, restrictocin and fumitremorgin B were not as toxic to the *Bom^Δleft* as to the *Bom^Δ55C* line (Fig. 6A–C), suggesting that the four remaining genes in *Bom^Δleft* are not sufficient to confer protection against restrictocin and fumitremorgin B and that at least some of the six genes removed by this deficiency are required for host defense against these mycotoxins. Note that in the experiments of direct injection of mycotoxins, we are likely delivering higher doses of mycotoxin into the host than those actually secreted by wild-type *A. fumigatus* during infection; this likely accounts for the apparent discrepancy between this approach and that of using mycotoxin-deficient *A. fumigatus* strains reported in the paragraph above.

We next injected restrictocin or verruculogen into the *BomT1-Gal4* KI and the *BomBc1* indel mutants. We found that both mutants are as sensitive to restrictocin as *MyD88* mutants, whereas none of them displayed any enhanced susceptibility to a verruculogen challenge (Fig. 6D–G).

Finally, as a proxy to analyze the function of *BomSs* in the resilience to *A. fumigatus* mycotoxins, we tested *Bbd* mutants in which no BomSs are detectable in the hemolymph (Lin et al, 2019). We found *Bbd* mutant flies to be as sensitive to *A. fumigatus* as *MyD88* flies (Appendix Fig. S12A). They were somewhat sensitive to restrictocin, highly sensitive to verruculogen and not susceptible to a gliotoxin challenge (Appendix Fig. S12B–D). Of note, we cannot formally exclude a contribution of *55C BomTs* to these phenotypes since it is unknown whether Bbd is required for the secretion/stability of BomTs in the hemolymph. We have previously documented the potential for *BomS3* (restrictocin only) and especially *BomS6* to rescue the sensitivity of *Bom^Δ55C* flies to injected restrictocin or verruculogen (Xu et al, 2023).

## Discussion

In this article, we report a genetic dissection of the *55C Bomanin* locus using both loss-of-function and overexpression strategies. A major outcome of this study is the identification of BomT1 as being a major effector of host resistance against *E. faecalis* infection, albeit it likely functions together with other gene product(s) of the locus in preventing *E. faecalis* proliferation. Interestingly, the same Bomanin peptide appeared to be required in the host defense against *M. robertsii* natural infection; we failed to obtain evidence supporting a role for it in resistance against this infection, that is, that it has any antifungal activity. Likewise, we found that *BomS2* but not *BomS5* null mutants are required for protection against injected *M. robertsii* spores, even though the fungal burden remained unaltered as compared to wild-type flies. Thus, both *BomT1* and *BomS2* may function in resilience against *M. robertsii*, but in distinct infection models. We also report that two Bomanins are required for protection against a ribotoxin secreted by *A. fumigatus*, BomBc1 and again BomT1, implying that this latter Bom peptide is involved both in resistance and resilience to infections. Finally, we have discovered that *C. albicans* dissemination and its lethality can be uncoupled, with again BomTs having the capacity to prevent to some extent the dissemination of *C. albicans*. Thus, the functions of *55C Bomanins* are complex and

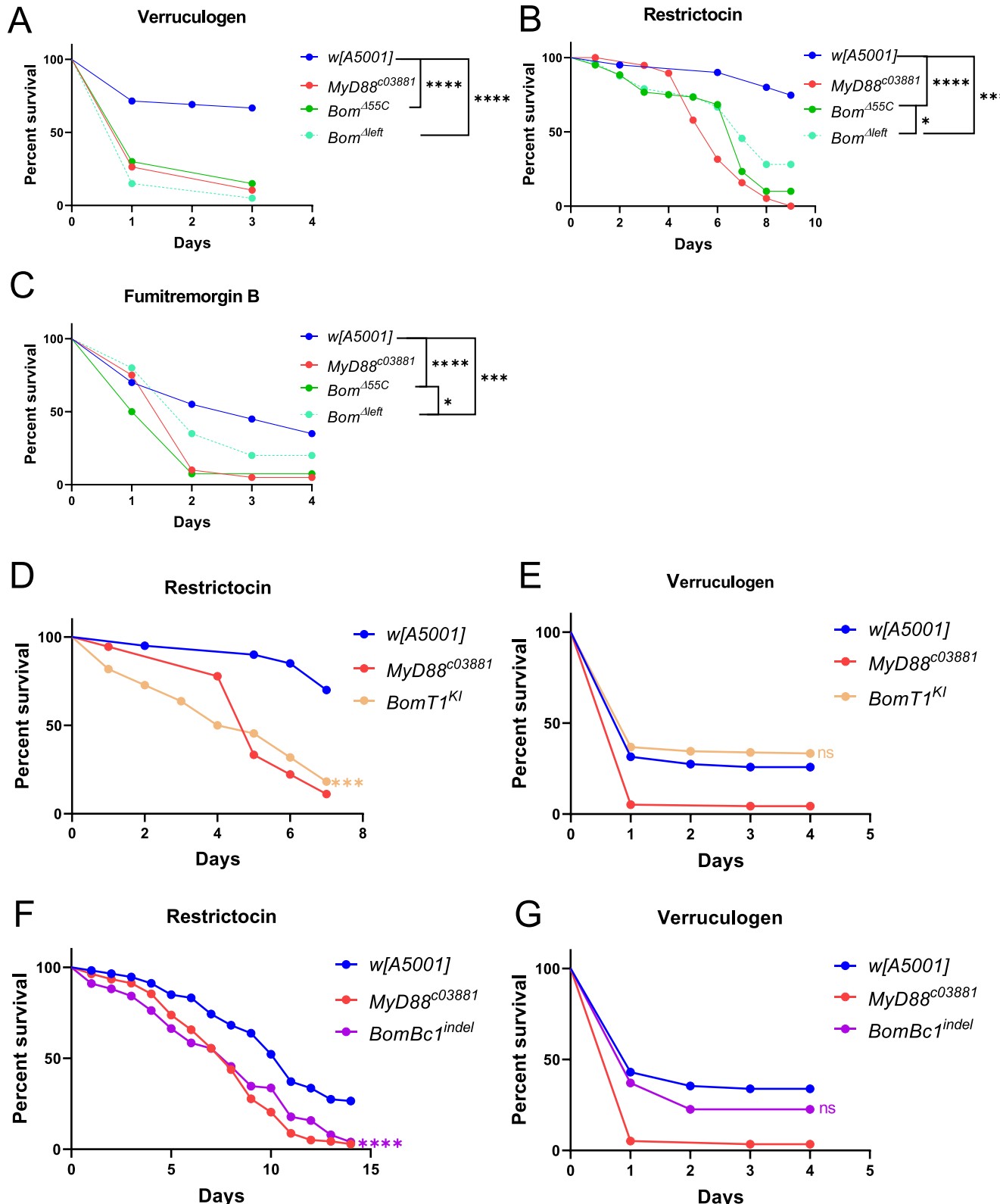

**Figure 6.  55C Bomanins confer protection against A. fumigatus mycotoxins.**

(A–C) Survival curves of Bom^ΔSSC, Bom^Δleft, MyD88^c03881, and w [A5001] flies after verruculogen (5 mg/ml, 4.6 nL/fly) (A), restrictocin (5 mg/ml, 2 nL/fly) (B), and fumitremorgin B (5 mg/ml, 2 nL/fly) (C) injection. (D, E) Survival curves of BomT1-Gal4 KI (BomT1^KI) flies after restrictocin (D) and verruculogen (E) injection. (F, G) Survival curves of BomBc^indel flies after restrictocin (F) and verruculogen (G) injection. Data information: In (A–G), three experiments were performed at different times. Except for (E, G) that used biological duplicates, each experiment used biological triplicates of 20 flies in parallel. Unless shown otherwise, the data were analyzed between infected-mutant and - w [A5001] fly using the Log-Rank test; ns, no significant difference. (A) Bom^Δleft and Bom^ΔSSC flies: P < 0.0001; (B) Bom^ΔSSC flies: P < 0.0001, Bom^Δleft flies: P = 0.0005, Bom^Δleft flies VS Bom^ΔSSC flies: P = 0.045; (C) Bom^ΔSSC flies: P < 0.0001, Bom^Δleft flies: P = 0.0009, Bom^Δleft flies VS Bom^ΔSSC flies: P = 0.017; (D) BomT1^KI: P = 0.0002; (E) BomT1^KI: P = 0.17; (F) BomBc^indel: P < 0.0001; (G) BomBc^indel: P = 0.19. Source data are available online for this figure.

intertwined and involved in both the resistance and resilience facets of host defense, with BomT1 playing a role in both aspects of host defense, depending on the infection model. We discuss below specific issues raised by this work and speculate at the end on a putative mechanism that may account for the various properties of this family of effector peptides.

## Limitations of the study

Even though we have generated four null and one hypomorphic mutants at the 55C locus, we have failed to obtain any mutant / RNAi line affecting the expression of BomS1, BomS3, and BomS6; we obtained data for BomS4 and BomBc2 only by silencing gene expression with RNAi lines. As regards BomS4, we have analyzed only one RNAi line, which however appears to be rather effective at silencing the already weakly expressed BomS4 transcripts (Appendix Fig. S7C). Yet, we cannot exclude a hypomorphic effect or a potential accumulation of the basally expressed BomS4 peptide in a specific cell type or tissue prior to the silencing of the transcripts at the adult stage. Thus, a null BomS4 mutant would be required to confirm its apparent absence of roles in the host defense against these bacterial and fungal pathogens.

Of interest, this study confirms that distinct host defenses, namely different Bomanin peptides, are relevant according to the route of infection of M. robertsii (Wang, 2020). Notwithstanding, another limitation is that we have tested here only a limited panel of bacterial, yeast, and filamentous fungal pathogens.

Even though the overexpression of Bom genes in Bom^ΔSSC mutants in the absence of immune challenge is not as strong as that induced by M. luteus in wild-type flies, except for BomT1, we report in this study some specificity in the effects of mildly overexpressed Bomanins, with some overlap in survival phenotypes to infections between some but not all Bomanins (Fig. 2A), in contrast to a previous study on the redundant functions against N. glabrata of strongly overexpressed BomS genes (Lindsay et al, 2018).

## Role of BomT1 in the resistance to E. faecalis infection

Clemmons et al (2015) had reported that the 55C locus is required for resistance against E. faecalis. We found that BomT1 mutants displayed a susceptibility to this bacterial pathogen that was almost as pronounced as that displayed by MyD88, suggesting that it is a major mediator of resistance to E. faecalis since the bacterial burden is increased in the BomT1 mutant (Fig. 3D). Yet, it may not act alone against this bacterium since we observed only a moderate rescue of the Bom^ΔSSC sensitivity phenotype by its overexpression (Figs. 2A and 3C). We note that Bom^Δleft are also as susceptible to this infectious challenge as Bom^ΔSSC flies, suggesting that the four

rightmost genes of the locus are not sufficient to mediate any protection on their own. Interestingly, BomS1 overexpression also somewhat rescues Bom^ΔSSC flies (Figs. 2A and EV2B) and it would be interesting to test whether their joint overexpression in this immunodeficient context would bring the protection to wild-type levels. Unexpectedly, we found that BomBc2 provided a degree of protection against E. faecalis when overexpressed in wild-type but not Bom^ΔSSC flies (Figs. 2A and EV2C), possibly reflecting a positive interaction with another 55C Bomanin gene product, possibly BomT1. Even though the two RNAi lines that target BomBc2 appear to be quite effective in silencing this gene, (Appendix Fig. S7C), BomBc2 null mutant will be required to clarify the function of this gene in host defense against infections. A weak phenotype of susceptibility to E. faecalis has previously been reported upon silencing BomBc1 but not BomT1, BomS1, and BomS4 (Chapman et al, 2020), in contrast to the results presented in our study (Fig. 2C).

## Involvement of 55C genes in resilience to infections

BomS2 behaves much as BomT1 mutants, with no altered M. robertsii fungal burden, respectively after injection of spores or natural infection, thus arguing against a role in resistance. It is an open possibility that BomT1 and BomS2 rather function in resilience to secreted virulence factors, which may be distinct in the two infection models (Wang, 2020). We have reported previously that some BaraA-derived peptides provide a degree of protection against Destruxin A, a toxic hexadepsipeptide secreted by the fungus (Huang et al, 2023). It remains unclear whether the 55 C Bomanin locus is involved in resilience to Destruxin A as we have obtained contrasted results in our experiments. However, the number of proteins secreted by M. robertsii, many of them proteases, and of secondary metabolites is so high that the two Bomanins may protect against virulence factors distinct from Destruxin A (Gao et al, 2011). Interestingly, some M. robertsii virulence factors able to inhibit the activation of the Toll pathway by either the GNBP3 sensor or the PSH protease bait have recently been identified (Lu et al, 2024; Tang et al, 2025). However, as pointed earlier (Clemmons et al, 2015), the basal expression levels of many Bomanins in adult fat body or carcass is high, especially those of BomS1, BomBc2, BomS2, BomS3, BomT2, and BomS6. This basal expression might be sufficient to provide a degree of protection despite the pathogen attack on the activation of the Toll pathway.

Nevertheless, the strongest evidence we have obtained for a role of Bomanins in resilience against a mycotoxin is the sensitivity of both BomT1 and BomBc1 to a challenge by the A. fumigatus mycotoxin restrictocin (this work).

Of note, a recent study has documented a role for *BomS2* in the host defense against *Lysinibacillus fusiformis*, a mild Gram-positive pathogen, the bacterial burden of which was not reported in *BomS2* mutants (Smith et al, 2023). With respect to *E. faecalis*, we did observe a mutant phenotype for only one of two null mutants (Figs. 2C and EV2A).

## The dissemination of *C. albicans* within the Drosophila body may not be essential for its pathogenicity

*C. albicans* is a dimorphic yeast able to form hyphae. The transition between the yeast and filamentous forms has been reported to be important for its virulence and it is believed that hyphae allow it to invade tissues thereby promoting its pathogenicity. Here, we find that *Bom$^{\Delta55C}$* mutants are highly permissive to the dissemination of the fungus, in keeping with its sensitivity to this infection. The rescue experiments of *Bom$^{\Delta55C}$* by single *Bom* gene overexpression revealed that *BomT1* and *BomT2* strongly prevent the dissemination of *C. albicans* within the fly yet do not allow it to better survive the infection. Since *BomT1* or *BomT2* overexpression did not protect *Bom$^{\Delta55C}$* mutants from *C. albicans* infection (Fig. 2A), these findings suggest that *C. albicans* can kill its host in the absence of a widespread dissemination. Thus, this pathogen may kill *Bom$^{\Delta55C}$* hosts through secreted virulence factors, a situation akin to that recently described for *A. fumigatus* infection (Xu et al, 2023). These factors may be distinct from candidalysin that appears to be secreted by hyphae in the invasion pocket of an epithelium (Mogavero et al, 2021). We note that the overexpression of the *w$^{1118}$* isoform of *BomBc1* is the only *Bom* gene able to protect *Bom$^{\Delta55C}$* flies from *C. albicans* infection (Fig. 2A). Even though it is slightly less efficient than BomTs at preventing the dissemination of *C. albicans* to the head or abdomen in *Bom$^{\Delta55C}$* mutant flies, the *w$^{1118}$* isoform of *BomBc1* does hinder the spreading of *C. albicans* to these tagmata distal to the injection point in the thorax (Fig. 5B). Interestingly, the silencing of *BomBc1* by one but not the other RNAi line leads to a sensitivity of the silenced flies to *C. albicans* infection (Fig. 2C), which highlights the potential role of *BomBc1* in host defense against this pathogenic dimorphic yeast.

Taken together, our data suggest that some long/tailed Bomanin peptides interfere with dissemination, possibly by affecting filamentation. Interestingly, Daisho peptides, which are related to Bomanins (Clemmons et al, 2015), have been shown to bind to hyphae of the mold *Fusarium oxysporum* against which they are active (Cohen et al, 2020).

This study reveals that distinct Bomanins have different activities in host defense against infections, ranging from antimicrobial activities against a prokaryote pathogen, *E. faecalis*, against eukaryotic pathogens including yeast and filamentous fungi and also provide protection against secreted virulence factors. Remarkably, *BomT1* plays a role in resistance to *E. faecalis*, a potential one in resilience against *M. robertsii* natural infection, can prevent the dissemination of *C. albicans*, and is required for protection against the *A. fumigatus* ribotoxin Restrictocin but not against the secondary metabolite verruculogen. Strikingly, BomS6 is able to protect flies from the action of a protein toxin, Restrictocin, that cleaves 28S RNA; it can also foster the recovery of *Bom$^{\Delta55C}$* from exposure to verruculogen, which targets a maxi-potassium channel, Slowpoke, by binding to a site within its transmembrane domain (Raisch et al, 2021; Xu et al, 2023). At first sight, it is

difficult to envision how this family of related peptides manages to function in such diverse facets of host defense. We would like to speculate here that there is a potential common target of this family of Toll pathway effectors, namely, cytoplasmic membranes. Like several AMPs such as Cecropins, antimicrobial Bomanins such as BomT1 may interact with the bacterial membrane and possibly directly lead to its lysis, likely in association with other factors such as other Bomanins. A similar process may be at play as regards the candidacidal activity of BomS or a potential action on hyphae of BomTs/BomBc1. With regard to the protection against the ribotoxin, we note that a remarkable feature of Restrictocin is its ability to cross membranes. Future studies will tell whether BomS6 is able to modify the host cell permeability to the ribotoxin and whether the modification of the biophysical properties of neural membranes induces a conformational change in the Slowpoke receptor that would mask its verruculogen binding site, thereby allowing the host to recover from tremors induced by this neuromycotoxin (Xu et al, 2023).

## Methods

### Reagents and tools table

| Reagent/resource | Reference or source | Identifier or catalog number |
|---|---|---|
| **Experimental models** | | |
| *CEA17ΔakuB$^{Ku80}$* | Jean-Paul Latge lab. | N/A |
| A1160 | Axel Brakhage lab. | N/A |
| *ΔftmA* | Xu et al, 2023 | N/A |
| *Δaspf1* | Xu et al, 2023 | N/A |
| *Micrococcus luteus* | China General Microbiological Culture Collection Center | CGMCC#1.2299 |
| *Enterococcus faecalis* | China General Microbiological Culture Collection Center | CGMCC#1.2135 |
| *Nakaseomyces glabrata (Candida glabrata)* | ATCC | ATCC2001 |
| *Candida albicans* | Malcolm Whiteway | CAM15.4 |
| *Metarhizium robertsii* | Chengshu Wang Lab. | 2575 |
| *D. melanogaster: MyD88$^{c03881}$* | Thibault et al, 2004 | c03881 |
| *D. melanogaster: Bom$^{\Delta55C}$* | Steven A Wasserman Lab. | N/A |
| *D. melanogaster: Bom$^{\Delta left}$* | Steven A Wasserman Lab. | N/A |
| *D. melanogaster: w$^{A5001}$* | Thibault et al, 2004 | N/A |
| *D. melanogaster:* Canton-S | Bloomington Drosophila Stock Center | BDSC64349 |
| *D. melanogaster: w$^{1118}$* | Vienna Drosophila Resource Center | VDRC60000 |
| *D. melanogaster:* Ubi-GAL4; UAS-BomS1 | This paper | N/A |
| *D. melanogaster:* Ubi-GAL4; UAS-BomS2 | This paper | N/A |
| *D. melanogaster:* Ubi-GAL4; UAS-BomS3 | This paper | N/A |

| Reagent/resource | Reference or source | Identifier or catalog number |
|---|---|---|
| D. melanogaster: Ubi-GAL4; UAS-BomS4 | This paper | N/A |
| D. melanogaster: Ubi-GAL4; UAS-BomS5 | This paper | N/A |
| D. melanogaster: Ubi-GAL4; UAS-BomS6 | This paper | N/A |
| D. melanogaster: Ubi-GAL4; UAS-BomT1 | This paper | N/A |
| D. melanogaster: Ubi-GAL4; UAS-BomT2 | This paper | N/A |
| D. melanogaster: Ubi-GAL4; UAS-BomBc1 | This paper | N/A |
| D. melanogaster: Ubi-GAL4; UAS-BomBc2 | This paper | N/A |
| D. melanogaster: Ubi-GAL4, MyD88/Cyo; UAS-BomS1 | This paper | N/A |
| D. melanogaster: Ubi-GAL4, MyD88/Cyo; UAS-BomS2 | This paper | N/A |
| D. melanogaster: Ubi-GAL4, MyD88/Cyo; UAS-BomS3 | This paper | N/A |
| D. melanogaster: Ubi-GAL4, MyD88/Cyo; UAS-BomS4 | This paper | N/A |
| D. melanogaster: Ubi-GAL4, MyD88/Cyo; UAS-BomS5 | This paper | N/A |
| D. melanogaster: Ubi-GAL4, MyD88/Cyo; UAS-BomS6 | This paper | N/A |
| D. melanogaster: Ubi-GAL4, MyD88/Cyo; UAS-BomT1 | This paper | N/A |
| D. melanogaster: Ubi-GAL4, MyD88/Cyo; UAS-BomT2 | This paper | N/A |
| D. melanogaster: Ubi-GAL4, MyD88/Cyo; UAS-BomBc1 | This paper | N/A |
| D. melanogaster: Ubi-GAL4, MyD88/Cyo; UAS-BomBc2 | This paper | N/A |
| D. melanogaster: Ubi-GAL4, Bom$^{\Delta55C}$/Cyo; UAS-BomS1 | This paper | N/A |
| D. melanogaster: Ubi-GAL4, Bom$^{\Delta55C}$/Cyo; UAS-BomS2 | This paper | N/A |
| D. melanogaster: Ubi-GAL4, Bom$^{\Delta55C}$/Cyo; UAS-BomS2 | This paper | N/A |
| D. melanogaster: Ubi-GAL4, Bom$^{\Delta55C}$/Cyo; UAS-BomS3 | This paper | N/A |
| D. melanogaster: Ubi-GAL4, Bom$^{\Delta55C}$/Cyo; UAS-BomS4 | This paper | N/A |
| D. melanogaster: Ubi-GAL4, Bom$^{\Delta55C}$/Cyo; UAS-BomS5 | This paper | N/A |
| D. melanogaster: Ubi-GAL4, Bom$^{\Delta55C}$/Cyo; UAS-BomS6 | This paper | N/A |
| D. melanogaster: Ubi-GAL4, Bom$^{\Delta55C}$/Cyo; UAS-BomT1 | This paper | N/A |
| D. melanogaster: Ubi-GAL4, Bom$^{\Delta55C}$/Cyo; UAS-BomT2 | This paper | N/A |
| D. melanogaster: Ubi-GAL4, Bom$^{\Delta55C}$/Cyo; UAS-BomBc1 | This paper | N/A |

| Reagent/resource | Reference or source | Identifier or catalog number |
|---|---|---|
| D. melanogaster: Ubi-GAL4, Bom$^{\Delta55C}$/Cyo; UAS-BomBc2 | This paper | N/A |
| BomT1 $^{KI}$ | Wellgenetics | N/A |
| BomT1 $^{KD}$ | Bloomington Drosophila Stock Center | BDSC42617 |
| BomBc1 $^{KD1}$ | Bloomington Drosophila Stock Center | BDSC65901 |
| BomBc1 $^{KD2}$ | Vienna Drosophila Resource Center | VDRC15384 |
| BomBc1 $^{Indel}$ | Guangzhou Drosophila Resource Center | 30-66-11 |
| BomS4 $^{KD}$ | TsingHua Fly Center | THU5656 |
| BomBc2 $^{KD1}$ | Vienna Drosophila Resource Center | VDRC13926 |
| BomBc2 $^{KD2}$ | Bloomington Drosophila Stock Center | BL61348 |
| BomS2 $^{KO6}$ | This paper | N/A |
| BomS2 $^{KO36}$ | This paper | N/A |
| BomT2 $^{KD}$ | Vienna Drosophila Resource Center | VDRC103059 |
| BomT2 $^{Indel}$ | SFHI (Sino-France Hoffman Institute) | N/A |
| BomT2 $^{KI312}$ | This paper | N/A |
| BomT2 $^{KO259}$ | This paper | N/A |
| BomS5 $^{KO}$ | This paper | N/A |
| **Recombinant DNA** | | |
| pGW-HA-attB-barcode sequence | This paper | N/A |
| pGW-HA.attB | E Furger and J Bischof | N/A |
| pDONR™221 | Life Technologies | cat. no. 12536-017 |
| pCFD5 | Fillip Port | N/A |
| pBluescript-SK+ | This paper | N/A |
| pUAST-mCherry | This paper | N/A |
| **Oligonucleotides and other sequence-based reagents** | | |
| Vector-specific primers | This paper | N/A |
| Transgene-specific primers | This paper | N/A |
| RT–qPCR primers | Xu et al, 2023 | N/A |
| Primer for gRNA plasmid | This paper | N/A |
| **Chemicals, enzymes, and other reagents** | | |
| pyrithiamine | Merck | P0256 |
| potato dextrose agar (PDA) medium | Huankai Microbio Tech | 021050 |
| Tryptic soy broth (TSB) medium | Sigma-Aldrich | T8907 |
| Luria-Bertani (LB) medium | Caisson | CAI-LBP03 |
| Brain-Heart infusion Broth-medium | Oxoid | CM1135T |
| MRS medium | Binder | AA304 |

| Reagent/resource | Reference or source | Identifier or catalog number |
|---|---|---|
| yeast peptone dextrose-agar (YPDA) medium | Coolaber | PM2011 |
| ampicillin | Beyotime | ST008 |
| tetracycline | MedChemExpress | HY-A0107 |
| chloramphenicol | Coolaber | CC3451 |
| erythromycin | Solarbio | IE0200 |
| kanamycin | Solarbio | K1030 |
| Tween20 | Solarbio | T8220 |
| Dimethyl sulfoxide (DMSO) | Sigma | D2650 |
| restrictocin | Sigma | R0389 |
| verruculogen | Abcam | ab141889 |
| fumitremorgin B | MCE | HY-117313 |
| UVITEX 2B | Polysciences | 19517-10 |
| phosphate buffer | Macklin | P885771 |
| glutaraldehyde | Sigma-Aldrich | G6403 |
| 4% Paraformaldehyde Fix Solution | Innochem | B42135 |
| Trizol Reagent | Invitrogen | 15596-018 |
| Halt™ Protease Inhibitor Cocktail EDTA-free | Thermo Scientific™ | 87785 |
| Sapphire Chips | AbD Serotec | C14012 |
| PerfeCTa® qPCR ToughMix® UNG | Apexbio | 95138-250 |
| EVAGREEN | Apexbio | AK200002 |
| SS III ONE-STEP HI FI 100 RXN | Invitrogen | 12574035 |
| PerfectStart Green qPCR SuperMix | TransGen Biotech | AQ601-01 |
| Latex beads | Sigma-Aldrich | LB8 |
| Sapphire Chips | AbD Serotec | C14012 |
| PerfeCTa® qPCR ToughMix® UNG | Apexbio | 95138-250 |
| EVAGREEN | Apexbio | AK200002 |
| SS III ONE-STEP HI FI 100 RXN | Invitrogen | 12574035 |
| XhoI | Invitrogen | ER0692 |
| HindIII | Invitrogen | ER0505 |
| Competent cells (DB3.1, ccdB) | Invitrogen | A10460 |
| PfuDNA polymerase | Invitrogen | 11304011 |
| Platinum™ SuperFi II DNA Polymerase | Invitrogen | 14966001 |
| BP Clonase™ II enzyme | Thermo Fisher Scientific | 11789020 |
| LR Clonase™ II Plus enzyme | Thermo Fisher Scientific | 11791100 |
| TFA (trifluoroacetic acid) | Sigma | T6508 |
| The 4-hydroxy-α-cyanocinnamic acid (4-HCCA) | Sigma | 28166-41-8 |

| Reagent/resource | Reference or source | Identifier or catalog number |
|---|---|---|
| **Software** | | |
| GraphPad Prism V8.0.1 | GraphPad Software | N/A |
| mMass | http://www.mmass.org | N/A |
| LAS X V4.7.0 | leica | N/A |
| ZEN-lite 2 | Zeiss | N/A |

## Culture of flies

Flies were maintained under controlled environmental conditions (25 °C, 65% RH) in standard incubators. The nutritional medium was prepared by combining the following components per production batch: 1.2 kg cornmeal (Priméal), 1.2 kg glucose (Tereos Syral), 1.5 kg yeast (Bio Springer), 90 g nipagin (VWR Chemicals) diluted into 350 mL ethanol (Sigma-Aldrich), 120 g agar–agar (Sobigel). Ultrapure water was added to obtain a 25 L batch of food.

## Indel mutants and RNAi lines

The following mutant lines were used: the $Bom^{\Delta55C}$ and $Bom^{\Delta left}$ lines were a kind gift of Prof. Steven Wasserman (Clemmons et al, 2015) and have been isogenized in the w[A5001] background. A $MyD88^{c03881}$ isogenized line was used (Thibault et al, 2004; Xu et al, 2023). $BomT2^{\Delta tail}$ and $BomBc1^{indel}$ were generously provided by Prof. Jiyong Liu (Guangzhou Drosophila Resource Center). The $BomT1$-Gal4 KI line was constructed by Wellgenetics (Taipei, Taiwan); other KO and the $BomT2$-mCherry KI mutants were constructed by us using CRISPR-Cas9 technology. All mutant strains were isogenized in a w[A5001] background (Thibault et al, 2004). The following knockdown strains obtained from the Bloomington, Vienna, and Tsinghua stock centers were utilized: $BomT1$ (BL42617); $BomBc1$ (BL65901); $BomBc2$ (VDRC13926) and $BomBc2$ (BL61348); $BomT2$ (VDRC103059); $BomS4$ (THU5656). All the RNAi lines were crossed to a w; pUbi-Gal4, pTub-Gal80^{ts} (BDSC30140) driver lines at 18 °C; hatched adults were placed at 29 °C for 5 days to induce RNAi expression.

## Guide RNA plasmid construction

To achieve high specificity and efficiency in gene targeting, guide RNAs (gRNAs) were designed for each Bomanin-like gene based on the methodology previously reported by Ni and colleagues (Gratz et al, 2014). Specifically, the protospacer adjacent motif (PAM) sequence (NGG) within the homology arms was removed to minimize off-target effects. The pCFD5 plasmid, a generous gift from Dr. Fillip Port, served as the backbone for gRNA expression. Following the established protocols from the Fillip Port group, a Gibson assembly cloning kit (Port and Bullock, 2016) was employed to generate plasmids containing two gRNA sequences targeting a single Bomanin gene. Primer sequences used for the construction of gRNA plasmids are provided in the Appendix Table S1.

## Donor plasmid construction

The donor plasmid was constructed by first digesting the pBlue-script-SK+ plasmid with restriction enzymes PstI and SpeI. The mCherry sequence was obtained from the pUAST-mCherry plasmid, which served as the template for cloning the fluorescent marker. For each gene, the left homology arm (LA) was flanked by a PstI-digested site, while the right homology arm (RA) was flanked by a SpeI-digested site. The donor plasmid was assembled to include the following components: (i) a left arm of ~0.8–1.0 kb, with the reverse primer positioned immediately upstream of the start codon (ATG); (ii) the mCherry sequence, starting at the ATG and terminating at the stop codon; and (iii) a right arm of ~1.0–1.5 kb, beginning immediately downstream of the stop codon. Genomic DNA was used as the template for PCR amplification of the homology arms to ensure perfect sequence homology. All assembly reactions were performed using the Gibson assembly cloning kit (Invitrogen).

## Microinjection and generation of homology knock-in flies

For each gene, the donor plasmid and the corresponding gRNA plasmid were co-injected into lig4 mutant embryos (lig4 line obtained from the Bloomington Drosophila Stock Center, BDSC58492). Following injection, the embryos were allowed to develop into adults, and the resulting F1 progeny were screened for successful integration of the donor plasmid via PCR and sequencing (Appendix Table S2). To establish stable homology knock-in stocks, the F1 flies were crossed to balancer flies (yw; BcG/CyO). The resulting homozygous or heterozygous knock-in lines were then maintained and characterized for further analysis.

Knock-out strains originate from failed knock-in generation. For the knockout strains, $BomS5^{\Delta KO}$, $BomS2^{\Delta KO6}$, and $BomS2^{\Delta KO36}$, we performed isogenization to homogenize their genetic backgrounds, specifically in the $w[A5001]$ genetic background.

## Transgenic lines

The transgenic lines expressing single *Bom* genes of the *55C* locus under the *pUAS-hsp70* promoter control were generated as described below (Bischof et al, 2014).

## Transgenic overexpression of single *Bom* genes in different genetic backgrounds

### Generation of barcoded plasmid libraries

To construct barcoded plasmid libraries, we initiated the process by transforming the pGW-HA.attB vector into Escherichia coli DH5α competent cells. Transformed cells were cultured in 500 mL of LB broth and incubated overnight at 37 °C with shaking (220 rpm). Plasmid DNA was extracted using a standard alkaline lysis mini-prep protocol. The purified pGW-HA.attB vector was then subjected to double digestion with XhoI and HindIII restriction enzymes (New England Biolabs, Ipswich, MA, USA) according to the manufacturer's instructions.

Barcode oligonucleotides were ligated into the digested vector using T4 DNA ligase (Promega, Madison, WI, USA), following the optimized conditions described in prior work (Bischof et al, 2013). Ligation reactions were assessed for efficiency and diversity as

previously reported, with the top-performing reactions selected for downstream analysis. On the following day, approximately 600 colonies were screened from ten transformation plates, yielding an estimated library diversity of 60,000 unique barcodes. This diversity metric was calculated by normalizing colony counts to the theoretical transformation efficiency of the *E. coli* strain used.

A barcode diversity exceeding 50-fold the number of intended open reading frame (ORF) clones is recommended to mitigate saturation effects during high-throughput screening. Given our experimental design to overexpress 17 target genes, the observed library diversity of 60,000 met this criterion, ensuring adequate representation for our functional genomics studies.

### Generation of Bomanin transgenic fly lines via Gateway cloning

1. Cloning of Bomanin Open Reading Frames (ORFs) into Gateway-Compatible Vectors. The ORFs of Bomanin genes were cloned into the donor vector pDONR™221 (Life Technologies, no.12536-017) using BP recombination catalyzed by BP Clonase™ II enzyme (Thermo Fisher Scientific) following the manufacturer's protocol. This step generated entry clones for each Bomanin gene. Subsequently, LR recombination, mediated by LR Clonase™ II Plus enzyme (Thermo Fisher Scientific), was employed to transfer the Bomanin ORFs into the pGW-HA.attB-barcoded destination vector, resulting in individual expression vectors for each Bomanin gene. All Gateway reactions were performed according to the manufacturer's optimized conditions, and successful recombination was confirmed by colony PCR using vector-specific primers (Appendix Table S3);

2. Expression Vector Design and Transgenic Fly Generation. Each Bomanin ORF in the destination vector was placed under the control of the UAS-hsp70 promoter to enable tissue-specific expression upon heat shock induction. Transgenic flies were generated by micro-injecting the expression vectors into embryos of the *Drosophila melanogaster* strain BL24749 (Bloomington Drosophila Stock Center), which harbors the attP docking site and ΦC31 integrase on the third chromosome. Injected embryos were reared to adulthood, and F0 males were individually crossed to virgin *yw* (yellow-white) females. F1 progeny were screened for red-eyed males, indicative of successful transgene integration, using a dissecting microscope (Leica M205 FA, Leica Microsystems, Wetzlar, Germany). Genomic DNA from red-eyed F1 males was subjected to wing-clip PCR with transgene-specific primers (Appendix Table S3) to confirm the presence of the UAS-Bomanin construct;

3. Balancer Crosses and Stock Establishment. Positive F1 males were crossed to virgin females of the third chromosome balancer strain w; TM3/TM6B (Bloomington Drosophila Stock Center). F2 progeny were selected based on red eyes (indicative of the UAS-Bomanin transgene) and the TM6B balancer markers. Heterozygous F2 males and virgins carrying both the transgene and the balancer were crossed to remove the balancer, generating stable stocks for each Bomanin gene. Transgene integrity was verified by PCR using primers flanking the UAS-Bomanin insertion site, and barcode sequence fidelity was confirmed by Sanger sequencing using vector-specific primers (Appendix Table S3). The original expression vector served as a positive control for all PCR and sequencing reactions.

## Overexpression in wild-type background

Following the generation of transgenic UAS fly lines, the initial experimental step involved the collection of virgin females from the ubiquitin promoter-driven Gal4 line (genotype: *w; pubi-Gal4, ptub-Gal80^ts*.). These driver virgins were subsequently crossed with males from the transgenic UAS strain at 18 °C. Parental flies were transferred to fresh vials every 3 days to maintain optimal mating conditions and minimize overcrowding effects.

Upon larval hatching, the progeny was shifted to 29 °C for a period of 5–7 days to induce Gal80^ts-mediated derepression of the UAS-driven transgene. Only female offspring were utilized in subsequent infection assays to ensure genetic and physiological uniformity. Overexpression of the correct *Bom* gene in the wild-type genetic background was confirmed via quantitative reverse transcription PCR (RT–qPCR).

## Overexpression in *MyD88* mutant background

To generate a *MyD88*-deficient genetic background for transgene overexpression, the *w; pubi-Gal4, ptub-Gal80^ts* transgenes were recombined onto a second chromosome carrying a *MyD88^c03881* null allele. Homozygosity of the recombined line was confirmed by two criteria: (1) absence of *Drosomycin* expression following *E. faecalis* infection (as assessed via RT–qPCR) and (2) female sterility. The recombined *MyD88*-driver line was then crossed to *MyD88/CyO; UAS-Bomanin* flies. Progeny was reared at 18 °C and shifted to 29 °C for 5–7 days post-hatching to induce Gal80ts-mediated transgene expression.

## Overexpression in the *Bom^ΔSSC*-deficiency background

To study Bomanin overexpression in a genomic-deficiency background, the *w; pubi-Gal4, ptub-Gal80^ts* transgenes were recombined onto a chromosome bearing the *Bom^ΔSSC* deletion (spanning the Bomanin locus). The recombined line was crossed to *Bom^ΔSSC/CyO; UAS-Bomanin* flies. Offspring were maintained at 18 °C and transferred to 29 °C for 5–7 days after hatching to activate transgene expression. Overexpression in the *Bom^ΔSSC* deletion background was quantified using mass spectrometry (MS) and RT–qPCR.

The transgenic flies were crossed to a *w; pUbi-Gal4, pTub-Gal80^ts* driver line, in a homozygous *MyD88, Bom^ΔSSC* mutant or *w^A5001* background. The expression of the transgenes was checked by RT–qPCR and when detectable by mass spectrometry analysis in *Bom^ΔSSC* background on collected hemolymph of single flies.

## Quantitative RT-PCR

Total RNA was extracted from the whole body of adult flies using TRIzol® reagent (Invitrogen), following the manufacturer's protocol. For each biological replicate, five age-matched female flies were pooled to minimize individual variability. RNA integrity was confirmed via NanoDrop. Complementary DNA (cDNA) was synthesized from 1 μg of total RNA using the iScript™ cDNA Synthesis Kit (Bio-Rad), with random hexamer primers, according to the manufacturer's instructions. The cDNA solution was diluted sixfold prior to its use for the qPCR reactions. The primers used in quantitative PCR are listed in Appendix Table S4.

## Mass spectrometry

Single fly hemolymph was collected from individual female flies and immediately transferred to 2 μL of 0.1% trifluoroacetic acid (TFA) solution on ice to prevent protein degradation and maintain sample integrity. The 4-hydroxy-α-cyanocinnamic acid (4-HCCA) sandwich spotting matrix method was employed for sample loading. Two distinct matrix solutions were prepared: Matrix 1: A saturated solution of 4-HCCA (20 mg/mL) in 100% acetone. Matrix 2: A solution containing 20 mg/mL 4-HCCA in a 2:1 acetonitrile/0.1% TFA mixture. For each hemolymph sample from a single fly, 0.6 μL of the sample was loaded onto the MALDI target plate. The sample was directly applied to a dried bed of 0.5 μL of Matrix 1, and the sample was overlaid with 0.4 μL of Matrix 2 to ensure optimal crystallization and ionization efficiency during MS analysis. After air drying under ambient condition, the prepared samples were analyzed using a MALDI-TOF mass spectrometer. The instrument was calibrated with standard peptide/protein mixtures to ensure accurate mass measurements. Data acquisition was performed in positive ion mode over a mass range of 800–20,000 Da, with a laser intensity optimized for the detection of small peptides and proteins present in the hemolymph.

## Microbial culture and infection

*Enterococcus faecalis*. *E. faecalis* strains (*CGMCC #1.2135*) was maintained on Luria-Bertani agar (LBA) plates stored at 4 °C for at most a month. For experimental replicates, single colonies were isolated under sterile conditions and inoculated into 5 mL LB broth. Cultures were incubated at 37 °C with shaking (220 rpm) for 6–8 h to reach mid-logarithmic phase ($OD_{600} = 0.6$–$0.8$, measured via Nanodrop spectrophotometer). Cells were collected by centrifugation (7500 rpm, 2 min, 4 °C) and resuspended in phosphate-buffered saline (PBS). This washing procedure was repeated three times to remove residual culture media. Final bacterial suspensions were adjusted to $OD_{600} = 0.1$ concentration, ensuring experimental consistency across biological replicates. 4.6 nL of standardized bacterial suspension was delivered per fly via thoracic microinjection.

*Candida albicans* (*CAM15.4*), a kind gift of Malcolm Whiteway (Alarco et al, 2004) and *N. glabrata* (*ATCC2001*) were maintained on yeast extract peptone dextrose (YPD) plates at room temperature after two rounds of propagation on agar plates at 29 °C after retrieval from the −80 °C freezer. Single colonies were taken using a sterile tungsten needle for infections, 3–5 days old adult flies were anesthetized and pricked in the thorax region with the needle containing fungal cells (colony pricking assay). Mock-infected controls were performed using needles dipped into sterile PBS.

*Metarhizium robertsii* (*2575*), a kind gift of Prof. Chengshu Wang, was propagated on Potato Dextrose Agar (PDA; BD Difco™) plates at 25 °C for 7–14 days to ensure optimal sporulation. Spores were collected by gently scraping plate surfaces with PBST (PBS + 0.01% Tween20) for injection experiments or sterile ddH$_2$O + 0.01% Tween20 for natural infection assays. Suspensions were filtered through Miracloth to remove mycelial debris and quantified using a hemocytometer. For injection experiments, 4.6 nL of standardized conidial suspension ($10^7$ spores/mL) was delivered per fly via thoracic microinjection. For natural infection,

5 mL of diluted conidial suspension ($10^5$ spores/mL) was applied to groups of 20 flies, which were subsequently dried on a filter under a vacuum aspiration.

*Aspergillus fumigatus*. *A. fumigatus* strain maintenance and infection procedures were performed as previously described (Xu et al, 2023).

*Micrococcus luteus*. *M. luteus* was cultivated in Luria-Bertani Broth at 37 °C under aerobic conditions for 24 h. Bacterial cells were pelleted by centrifugation at 3000 rpm for 10 min, followed by two successive washing cycles in phosphate-buffered saline (PBS). The pellet was resuspended in 1 mL PBS after each centrifugation step to ensure complete removal of culture medium components. Needles were dipped in a concentrated pellet prior to pricking flies.

## Bacterial/fungal load in single flies

Individual infected flies were processed for microbial enumeration using a modified protocol as follows: single flies were transferred to 1.5 mL microcentrifuge tubes containing 20 μL PBS with Tween20 (PBST) and two 3 mm stainless steel beads. Samples were homogenized by mechanical disruption using a Mixer Mill (Retsch MM 300) at 30 Hz for 2 min. Homogenates were subjected to serial dilutions ($10^{-1}$ to $10^{-5}$) in PBST based on pre-determined time point-specific infection kinetics. Aliquots (50 μL) of each dilution were spread-plated in duplicate onto PDA plates for fungi or LBA ones for bacteria, with one fly equivalent per plate. Plates were incubated at organism-specific temperatures (29 °C for fungi, 37 °C for bacteria) in a humidified incubator. Fungal colonies were enumerated after 48 h incubation, while bacterial colonies were counted after 24 h. The limit of detection was established at 10 CFU/fly based on the lowest dilution yielding ≥30 colonies per plate. To measure the microbial burden at death, the flies were monitored regularly and collected individually within 30 min after their death.

## Mycotoxins preparation and injection

Mycotoxin standard solutions were prepared as described previously (Xu et al, 2023). Commercially available mycotoxin powders (restrictocin in PBS, verruculogen in dimethyl sulfoxide (DMSO), and fumitremorgin B in DMSO) were dissolved to generate stock solutions. Working concentrations were prepared by serial dilution in PBS or DMSO. Stock solutions were stored at −20 °C in amber vials protected from light, while fresh working solutions were prepared daily.

## Survival tests

In all, 5–7-day-old female adults were utilized to ensure uniform physiological maturity. Flies were housed in vials containing standard cornmeal-agar medium at a density of 20 individuals per vial, with three biological replicates maintained for each experimental condition. During infection studies, survival rates were monitored daily to record mortality and surviving flies were transferred to fresh vials every 3 days.

## In vivo phagocytosis

In total, 6000 live yeast cells of *N. glabrata* in a 0.01% Tween20 solution in PBS were injected into flies. Two hours after injection, a hole was made in the abdomen with tweezers. Using a long capillary adapted to a syringe after a flexible tube, 20 μL of PBS was injected in the thorax of each fly: the liquid going out from the opened abdomen was collected in a well from an eight-well microscope slide. Each well was previously filled with 20 μL of PBS to prevent desiccation. The samples were fixed in a 1% solution of paraformaldehyde in PBS for 10 min, kept in a humid chamber. After two washes in PBS for 5 min, the samples were blocked for 30 min in a PBS solution with 2% BSA. Extracellular yeasts and intracellular yeasts were then stained with primary antibody raised against whole UV-killed *Candida* (a kind gift from Prof. Karl Kuchler, Vienna), FITC-labeled and Cy3-labeled secondary antibody were used separately, respectively before (primary and FITC-labeled secondary antibody) and after a 30-min permeabilization of the hemocytes in a PBS solution containing 0.1% Triton X-100 and 2% bovine serum albumin (primary and Cy3-labeled secondary antibody). After the staining, with two washes in PBS for 5 min, the slide was mounted in Vectashield with DAPI. The samples were analyzed using a Zeiss Axioscope 2 fluorescent microscope (Liegeois et al, 2020). The number of red fluorescent bacteria that were not also green (phagocytosed bacteria) was determined for each DAPI-positive hemocyte, and the phagocytosis index was calculated (percentage of phagocytes containing at least 1 only-green bacterium) × (mean number of only-green bacteria per phagocyte).

## Analysis of *C. albicans* dissemination throughout the fly body

A sharp sterile needle was plunged into a single colony of CAM15.4 GFP-labeled *C. albicans* strain on a YPD plate and next used to wound *Drosophila* females in the middle of the thorax. The flies were incubated for 24 h and examined live with a SYCOP3 dissecting microscope with epifluorescence equipment (Zeiss) at maximum magnification of 112 using the FITC filter set and the different tagmata were examined at different focal planes for each individual fly and scored for the detection of a green signal (no such signal was observed in uninfected control flies). Pictures were taken using a AxioZoom.V16 camera (Zeiss) and visualized using the ZEN-lite 2 software (Zeiss).

## Statistical analysis and reproducibility

All survival data were analyzed using GraphPad Prism 9.0 software (GraphPad Software, San Diego, CA, USA). Survival curves were compared using the log-rank (Mantel–Cox) test unless otherwise specified in the figure legends. For quantitative analyses of microbial burden, gene expression levels, and GFP fluorescence intensity, unpaired, nonparametric Mann–Whitney statistical tests or Student two-tailed $t$ test were employed as appropriate for comparisons involving two samples; for multiple comparisons, Kruskal–Wallis test with Dunn's post hoc test or one-way ANOVA with Tukey's post hoc test were used as appropriate. Dissemination data were analyzed using m. Significance values: *$P < 0.05$; **$P < 0.01$; ***$P < 0.0003$; ****$P < 0.0001$; ns, not significant.

# Data availability

The raw data of the MALDI-TOF analysis (Appendix Fig. S9) were deposited on FigShare: https://figshare.com/s/8592656610278a376cd0.

The source data of this paper are collected in the following database record: biostudies:S-SCDT-10_1038-S44319-025-00559-6.

## Peer review information

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

## Acknowledgements

We thank Anne Beauvais and Jean-Paul Latge for the *A. fumigatus* strain used in this study, Chengshu Wang for the *M. robertsii* strain, Malcolm Whiteway for the GFP-*C. albicans* strain, Fillip Port for the gift of the pCFD5 plasmid, Karl Kuchler for the *Candida* antibody, Bruno Lemaitre, Jiyong Liu, Steven Wasserman, and the Guangzhou Drosophila Resource Center for fly stocks. Stocks obtained from the Bloomington Drosophila Stock Center, Vienna, and Tsinghua stock centers were also used in this study. We gratefully acknowledge the contributions of Miriam Yamba for expert technical help. YL was respectively partially funded through the Sino-Foreign cooperative graduate education project of Guangzhou Medical University, the International Training Plan for young outstanding scientific research talents of Guangdong Province, and the Overseas Postdoctoral Talent Support Program of Guangdong Province. This work was supported by the Association Platform BioPark of Archamps on its Research & Development budget (PB), from the National Natural Science Foundation of China Project (#32370931), the China High-end Foreign Talent Program, and the 111 Project (#D18010; China) to DF.

## Author contributions

**Yanyan Lou**: Conceptualization; Resources; Formal analysis; Validation; Investigation; Methodology; Writing—original draft; Writing—review and editing. **Bo Zhang**: Conceptualization; Resources; Formal analysis; Investigation; Methodology; Writing—original draft; Writing—review and editing. **Zhiyuan Zhang**: Resources; Investigation; Writing—review and editing. **Yingyi Pan**: Resources; Investigation; Writing—review and editing. **Jianwen Yang**: Resources; Investigation; Writing—review and editing. **Lu Li**: Resources; Investigation; Writing—review and editing. **Jianqiong Huang**: Resources; Investigation; Writing—review and editing. **Zihang Yuan**: Investigation; Writing—original draft; Writing—review and editing. **Samuel Liegeois**: Resources; Formal analysis; Validation; Investigation; Methodology. **Philippe Bulet**: Resources; Formal analysis; Validation; Investigation; Methodology. **Rui Xu**: Conceptualization; Funding acquisition. **Li Zi**: Conceptualization; Resources; Funding acquisition. **Dominique Ferrandon**: Conceptualization; Resources; Supervision; Funding acquisition; Validation; Methodology; Writing—original draft; Writing—review and editing.

Source data underlying figure panels in this paper may have individual authorship assigned. Where available, figure panel/source data authorship is listed in the following database record: biostudies:S-SCDT-10_1038-S44319-025-00559-6.

## Disclosure and competing interests statement

The authors declare no competing interests.

# Expanded View Figures

## Phagocytic index
## (2h after *N. glabrata* injection)

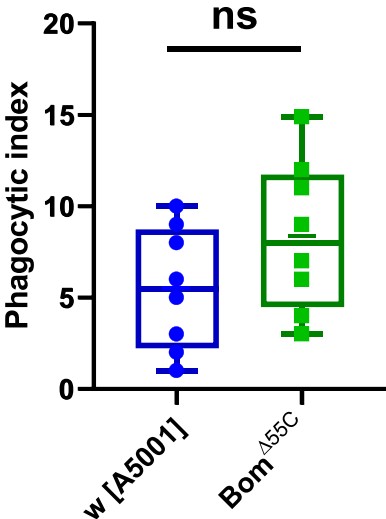

**Figure EV1.** ***Bomanins* at *55C* locus are not involved in phagocytosis.**

6000 live yeasts *N. glabrata* cells were injected into *w* [A5001] wild-type flies and *Bom*[ΔSSC] flies and their phagocytic index was monitored 2 h after injection. Data information: Mann–Whitney test. ns: not significant. The middle bar of the boxplot represents the median and the upper and lower limits of the box indicate respectively the first and third quartiles; the whiskers define the minima and maxima; *Bom*[ΔSSC] flies: $P = 0.15$. Source data are available online for this figure.

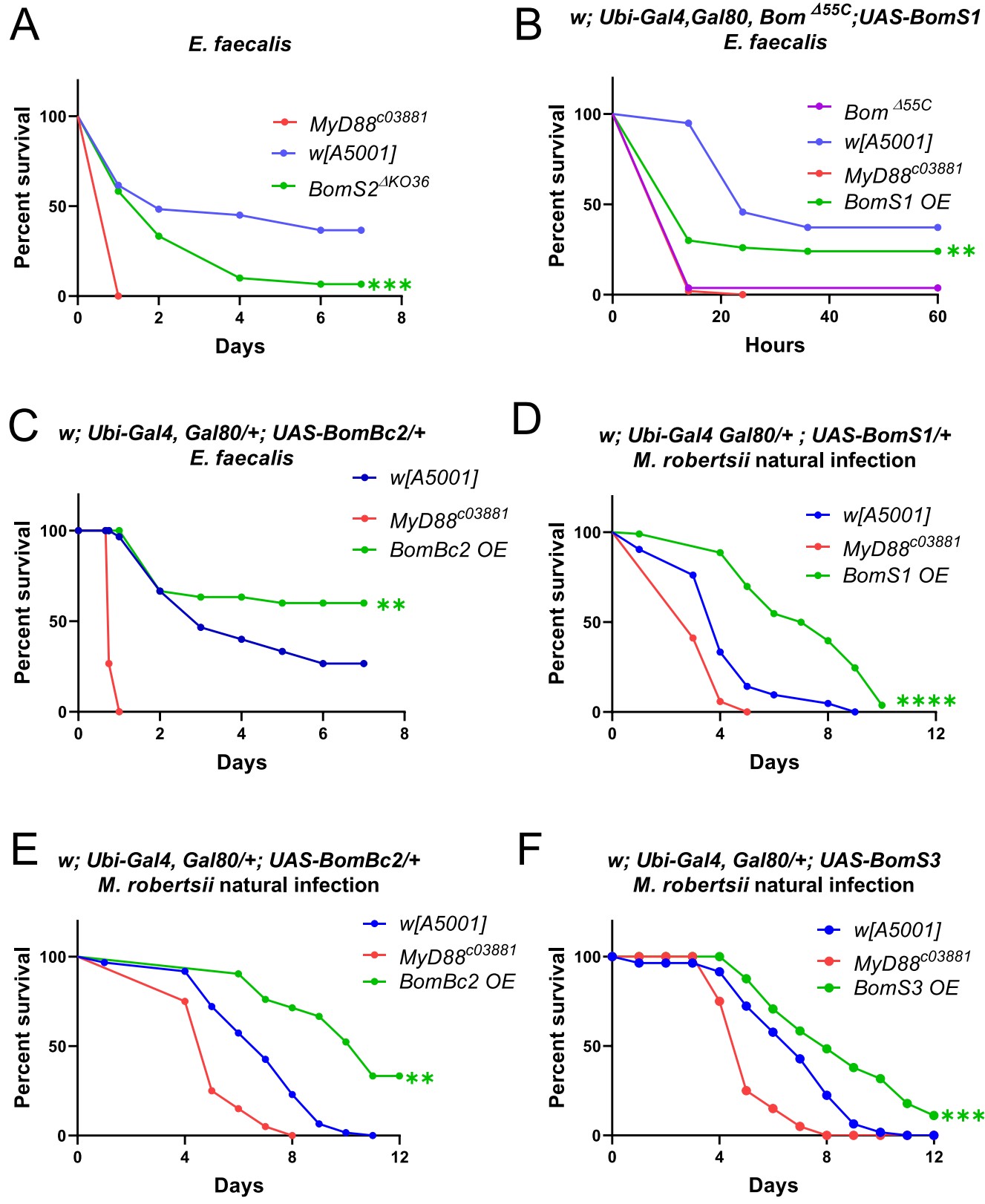

**Figure EV2.    Overexpression of *BomS1, BomBc2, or BomS3* in wild-type or *Bom^ΔSSC* background enhances resistance to *E. faecalis* or *M. robertsii*.**

(A) Survival curves of *BomS2^ΔKO36* null mutant flies after *E. faecalis* infection. (B) Survival curves of *BomS1*-overexpressing (OE) flies in a *Bom^ΔSSC* background after *E. faecalis* injection. (C) Survival experiments of *BomBc2*-overexpression (OE) flies in a wild-type background after *E. faecalis* injection. (D–F) Survival curves of *BomS1* (D), *BomBc2* (E)*, BomS3* (F) overexpressing (OE) flies in a wild-type background after *M. robertsii* natural infection. Data information: In (A–F) three experiments were performed at different times and each experiment used biological triplicates of 20 flies in parallel. The pooled data were analyzed between infected-mutant and - *w* [A5001] fly, except for (B) that compared *BomS1 OE* in a *Bom^ΔSSC* background to *Bom^ΔSSC*, using the Log-Rank test; ns, no significant difference. (A) *BomS2^ΔKO36* flies: $P = 0.0002$; (B) BomS1 OE: $P = 0.0013$; (C) BomBc2 OE: $P = 0.0007$; (D) BomS1 OE: $P < 0.0001$; (E) BomBc2 OE: $P = 0.0025$; (F) BomS3 OE: $P = 0.0009$. Source data are available online for this figure.

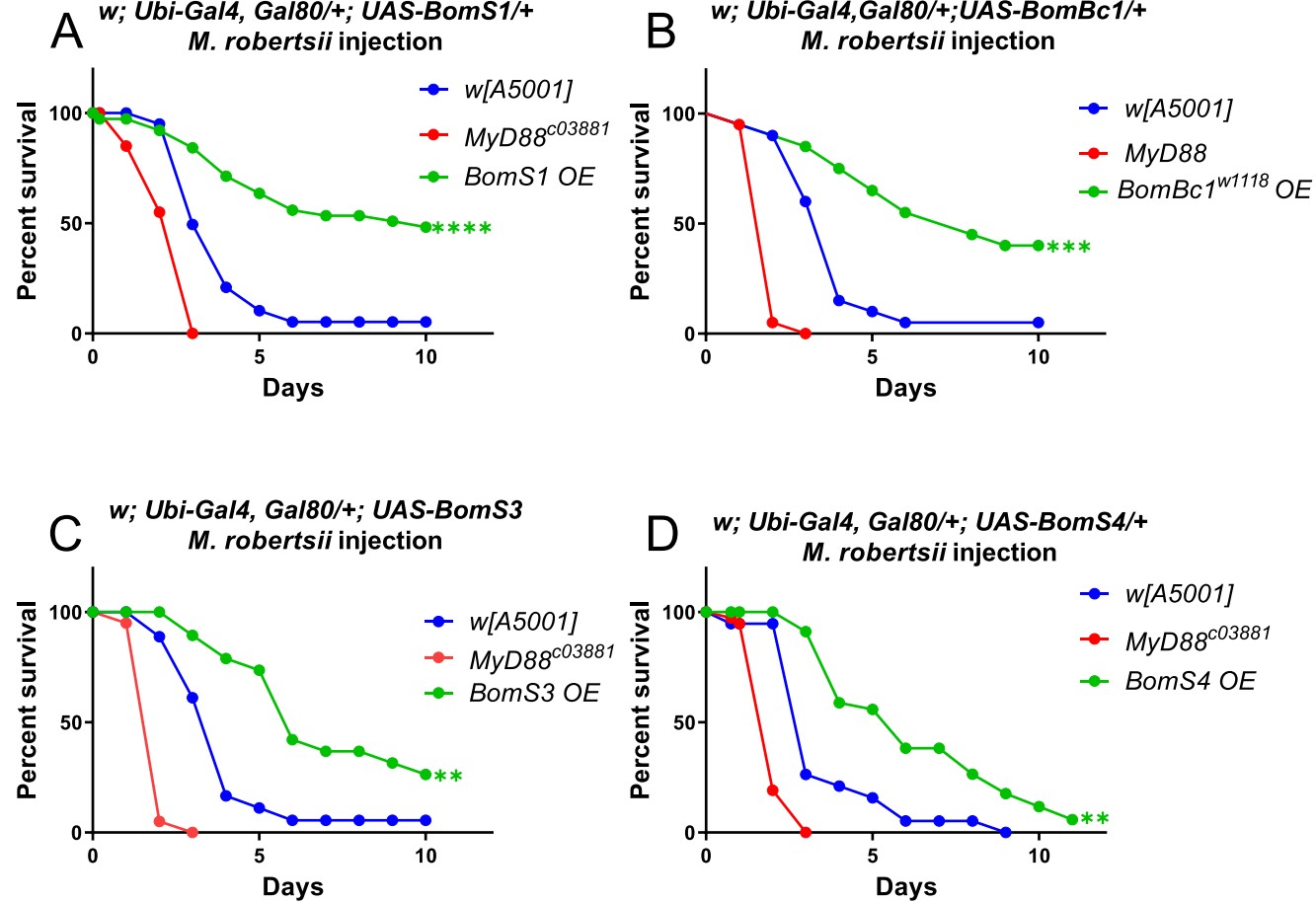

**Figure EV3. Overexpression of *BomBc1*, *BomS1*, *BomS3*, and *BomS4* in a wild-type genetic background can protect them from injected *M. robertsii* spores.**

(A–D) Survival curves of *BomBc1* (**A**), *BomS1* (**B**), *BomS3* (**C**), and *BomS4* (**D**)-overexpressing (OE) flies in a wild-type background after *M. robertsii* injection. Data information: three experiments were performed at different times and each experiment used biological triplicates of 20 flies in parallel. The pooled data were analyzed between infected-mutant and - *w* [A5001] fly using the Log-Rank test; ns, no significant difference. (**A**) BomS1 OE: $P < 0.0001$; (**B**) BomBc1w1118 OE: $P = 0.0003$; (**C**) BomS3 OE: $P = 0.0029$; (**D**) BomS4 OE: $P = 0.0072$. Source data are available online for this figure.

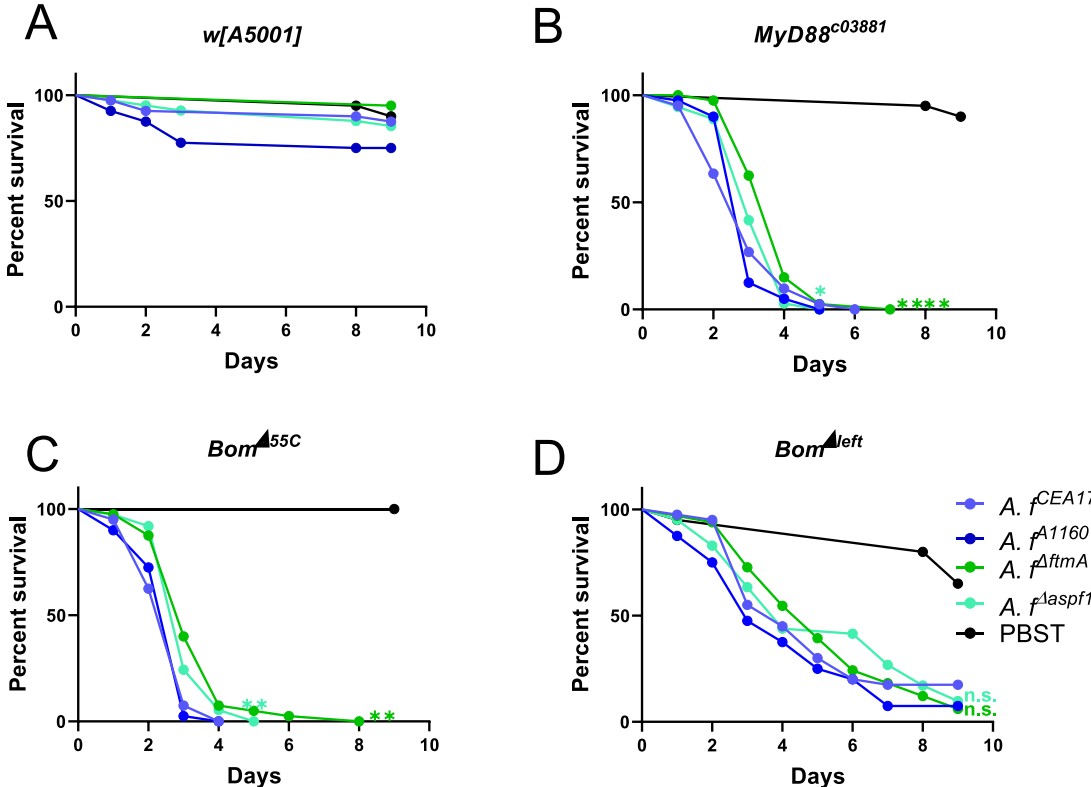

**Figure EV4. Mycotoxins mutants of *A. fumigatus* are as virulent as wild-type fungi when injected into *Bom^Δleft* flies.**

(A–D) Survival curves of *w[A5001]* (A), *MyD88^c03881* (B), *Bom^Δ55C* (C), and *Bom^Δleft* (D) flies to *A. fumigatus^ΔftmA* (fumitremorgin/verruculogen pathway inactivated), *A. fumigatus^Δaspf1* (restrictocin mutant), and *A. fumigatus* genetic background controls (*A. fumigatus^CEA17* or *A. fumigatus^A1160*) injection. Data information: three experiments were performed at different times and each experiment used biological triplicates of 20 flies in parallel. The pooled data were analyzed between mycotoxin mutant infected flies and wild-type *A. fumigatus* control-infected flies using the Log-Rank test; ns, no significant difference. (A) *A. fumigatus^ΔftmA*: $P = 0.22$, *A. fumigatus^Δaspf1*: $P = 0.86$; (B) *A. fumigatus^ΔftmA*: $P < 0.0001$, *A. fumigatus^Δaspf1*: $P = 0.047$; (C) *A. fumigatus^ΔftmA*: $P = 0.0013$; *A. fumigatus^ΔftmA*: $P = 0.0011$; (C) *A. fumigatus^ΔftmA*: $P = 0.93$; *A. fumigatus^ΔftmA*: $P = 0.99$. Source data are available online for this figure.

