## [Peer Review File · EMBO Reports]

Distinct Bomanins at the *Drosophila* 55C locus function in resistance and resilience to infections

Yanyan Lou, Bo Zhang, Zhiyuan Zhang, Yingyi Pan, Jianwen Yang, Lu Li, Jianqiong Huang, Zihang Yuan, Samuel Liégeois, Philippe Bulet, Rui Xu, Li Zi, and Dominique Ferrandon

Corresponding author(s): Dominique Ferrandon (D.Ferrandon@ibmc-cnrs.unistra.fr)

Review Timeline:

Submission Date:	16th Apr 25
Editorial Decision:	19th May 25
Revision Received:	24th Jun 25
Editorial Decision:	21st Jul 25
Revision Received:	5th Aug 25
Accepted:	11th Aug 25

Editor: Achim Breiling

Transaction Report:

Dear Dr. Ferrandon,

Thank you for the submission of your manuscript to EMBO reports. I have now received the reports from the three referees that were asked to evaluate your study, which can be found at the end of this email.

As you will see, the referees think that these findings are of interest. However, they have several comments, concerns, and suggestions, indicating that a major revision of the manuscript is necessary to allow publication of the study in EMBO reports. As the reports are below, and all the referee concerns need to be addressed, I will not detail them here.

Given the constructive referee comments, I would like to invite you to revise your manuscript with the understanding that the concerns of the referees must be addressed in the revised manuscript and/or in a detailed point-by-point response. Acceptance of your manuscript will depend on a positive outcome of a second round of review. It is EMBO reports policy to allow a single round of revision only and acceptance of the manuscript will therefore depend on the completeness of your responses included in the next, final version of the manuscript.

- 1) a .docx formatted version of the final manuscript text (including legends for main figures, EV figures and tables), but without the figures included. Figure legends should be compiled at the end of the manuscript text.
- 2) individual production quality figure files as .eps, .tif, .jpg (one file per figure), of main figures and EV figures. Please upload these as separate, individual files upon re-submission.

- 4) a complete author checklist, which you can download from our author guidelines (<https://www.embopress.org/page/journal/14693178/authorguide>). Please insert page numbers in the checklist to indicate where the requested information can be found in the manuscript. The completed author checklist will also be part of the RPF.

- 5) that primary datasets produced in this study (e.g. RNA-seq, ChIP-seq, structural and array data) are deposited in an

appropriate public database. If no primary datasets have been deposited, please also state this in a dedicated section (e.g. 'No primary datasets have been generated and deposited'), see below.

The accession numbers and database should be listed in a formal "Data Availability" section that follows the model below. This is now mandatory (like the COI statement). Please note that the Data Availability Section is restricted to new primary data that are part of this study. This section is mandatory. As indicated above, if no primary datasets have been deposited, please state this in this section

Data availability

6) We now request the publication of original source data with the aim of making primary data more accessible and transparent to the reader. You will receive a separate email with instructions for providing source data with your revised manuscript, including information how to upload and organize the files.

8) Regarding data quantification and statistics, please make sure that the number "n" for how many independent experiments were performed, their nature (biological versus technical replicates), the bars and error bars (e.g. SEM, SD) and the test used to calculate p-values is indicated in the respective figure legends (also for EV and Appendix figures). Please also check that all the p-values are explained in the legend, and that these fit to those shown in the figure. Please provide statistical testing where applicable. Please avoid the phrase 'independent experiment', but clearly state if these were biological or technical replicates. Please also indicate (e.g. with n.s.) if testing was performed, but the differences are not significant. In case n=2, please show the data as separate datapoints without error bars and statistics. See also: <http://www.embopress.org/page/journal/14693178/authorguide#statisticalanalysis>

9) Please add scale bars of similar style and thickness to microscopic images, using clearly visible black or white bars (depending on the background). Please place these in the lower right corner of the images themselves. Please do not write on or near the bars in the image but define the size in the respective figure legend.

10) Please also note our reference format:

12) We now use CRedit to specify the contributions of each author in the journal submission system. CRedit replaces the author contribution section. Please use the free text box to provide more detailed descriptions and do NOT provide your final manuscript text file with an author contributions section. See also our guide to authors: <https://www.embopress.org/page/journal/14693178/authorguide#authorshipguidelines>

13) All Materials and Methods need to be described in the main text using our 'Structured Methods' format, which is required for

all research articles. According to this format, the Methods section should include a Reagents and Tools Table (listing key reagents, experimental models, software, and relevant equipment and including their sources and relevant identifiers), uploaded as separate file, and a Methods section in which we encourage the authors to describe their methods using a step-by-step protocol format with bullet points, to facilitate the adoption of the methodologies across labs. More information on how to adhere to this format as well as downloadable templates (.doc) for the Reagents and Tools Table can be found in our author guidelines (section 'Structured Methods'):

14) Please add up to five keywords to the manuscript and order the sections like this, using these names:
Title page - Abstract - Keywords - Introduction - Results - Discussion - Methods - Data availability section - Acknowledgements - Disclosure and Competing Interests Statement - References - Figure legends - Expanded View Figure legends

15) Please make sure that all the funding information is also entered into the online submission system and that it is complete and similar to the one in the acknowledgement section of the manuscript text file.

I look forward to seeing a revised form of your manuscript when it is ready.

Please use this link to submit your revision: <https://embor.msubmit.net/cgi-bin/main.plex>

Yours sincerely,

Achim Breiling
EditorSenior
EMBO Reports

Referee #1:

This manuscript represents a valuable contribution to our understanding of the role of a recently described family of peptides, the bomanins, in the antimicrobial defence of *Drosophila*. The first members of this family turned up in an early screen for immune-induced peptides in *Drosophila*, carried out by Philippe Bulet and his coworkers (Uttenweiler-Joseph et al., 1998), but a detailed study of the entire family of bomanin peptides came much later (Clemmons et al., 2015, Lindsay et al 2018). The latter authors showed that a deletion of 10 of the 12 bomanins dramatically compromised the resistance against selected bacterial as well as fungal infections. They could also restore resistance by expressing some of the Bom genes in the deficient background, and thereby getting a first indication of the relative roles of some of the peptides. Taking a slightly different approach, the authors have now addressed the same problem, using null mutants, RNAi constructs, or by overexpressing single Bom genes and testing them for resistance against a panel of microorganisms.

It should be said that this is not an easy project; to identify the individual roles for twelve different but related genes, which may have overlapping functions and which may potentially also act in concert with each other in different constellations and against different microorganisms. Therefore it is not surprising that the authors are still far from achieving that goal. Nevertheless, they have showed convincing effects for some of them, especially regarding the survival of infected animals. They have also studied the effects on the dissemination of microorganisms in the animal and the resistance against fungal toxins.

The many pleiotropic effects of the different Bom genes, affecting survival or (uncorrelated) microbe dispersion or resistance to fungal toxins is a little disturbing for small molecules like the Bomanins. The results would be easier to interpret if we had a molecular mechanism, but that would be a much bigger question and outside the scope of this communication. The authors have also recognised this problem, and they have added some speculations about it in the Discussion.

Thus, while this may not represent a breakthrough in the study of these enigmatic peptides, this manuscript is an important step forward.

However, as described below, there are still a number of questions regarding the presentation of data.

Major issues:

1. Table 1 looks rather qualitative and it is very difficult to understand. The text states that "a key to understanding Table 1 is provided in Fig. EV2", but I have not found any of the "Expanded View" (EV) figures in the package I received.
2. Similarly, the data in Table 2 seem qualitative. What, exactly, do the sensitivity scores (++ or +++) mean, and what is the level of significance. Survival curves are more informative. Were they perhaps also provided in the missing "Expanded View" figures?
3. I did not understand the comment about the MS peaks in the section about "Genetic strategies..":
"MS being semi-quantitative, it was difficult to assess the overexpression of the gene products in the context of an immune response in wild-type flies; in addition, no ectopic expression was detected in the absence of an immune challenge in flies overexpressing BomS genes"
Could that be rephrased?
4. Related to the point above, Figure S9A is difficult for me to interpret. I assume that the arrows point to the specific peaks of the overexpressed peptides, but in some cases their positions seem to coincide with (background?) peaks in samples where that peptide should not be expressed. Have I misunderstood?
5. In the micrographs shown in Fig. 4A, the distribution (dissemination) of *Candida* cells is difficult to distinguish from other fluorescent structures in the background.
6. Maybe a question of editorial policy, but I think that important conclusions in the text should never rest on data that are shown only in supplementary figures or extended data files. The average impatient reader (like myself) may altogether skip the distracting and time-consuming step of switching back and forth between different documents while reading, and thereby miss the critical assessment of the strength of arguments.

Minor questions:

7. I have a problem with the commonplace generalisations about "Gram-positive" and "Gram-negative" bacteria, which are here faithfully echoed in the Introduction as well as elsewhere in this text. The Toll pathway, which is the major or only route for induction of Bomanin genes, is particularly responsive to bacterial peptidoglycans of the Lys type. Contrary to what is often assumed, such peptidoglycans are not ubiquitous among bacteria that belong to the (now rather obsolete) class of "Gram-positive" bacteria. In fact, many of the "Gram-positive" *Bacillus* and *Lactobacillus* species that are associated with insects have DAP-type peptidoglycans. This does not affect any of the conclusions in this text, but these self-perpetuating generalisations keep annoying me.
8. Please stick to standard nomenclature for mutants and other genetic constructs, with the genetic symbol (e.g. BomT1, MyD88) with a superscript for the mutant allele or other genetic variant. Here it is not always obvious if BomT1 (MyD88...) refers to a null mutant or an overexpression construct. Specifically, which MyD88 mutant was used here?
9. In the legend to Fig. S8, please explain "UC" (unchallenged?) , and "CS" (Canton S background?).
10. For the bacterial counts in dead flies it was not clear to me when they were sampled. I did not understand the explanation in the Material and Methods section: "Flies died in 30 minutes after injection". They must have been collected later than 30 minutes after injection, since the numbers e.g in Fig. 2E were in the range 5-25 million bacteria, much fewer than the maximum 4000 cells seen 2 hours after infection in living flies.
11. Related to the point above, the base 2 logarithmic scale is not very intuitive to most of us. Base 10 logarithms are better, perhaps with the actual number specified as in Fig. 2G.
12. Furthermore, the introduction of an acronym ("BLUD") for something as simple as the number of bacteria in dead flies seems unnecessary. Even more unnecessary is the additional acronym ("FLUD") for fungi. In general, acronyms should be used sparingly.
13. Apparently, "*C. glabrata*" is not closely related to *C. albicans* and is far from the genus *Candida*. Therefore, it is now renamed *Nakaseomyces glabratus* (*N. glabratus*). To avoid any misunderstanding about its relationship to *C. albicans*, please stick to the modern name. Thus, when it is first introduced, write:
"...*Nakaseomyces glabratus* (previously called *Candida glabrata*)...", and thereafter "*N. glabratus*".
14. In the first paragraph of the Discussion, you may have to explain the distinction between "the host defense against *M. robertsii* natural infection" and the "resistance against this infection".

Referee #2:

The authors have set out to study the individual Bomanins in the Bomanin cluster that contains 10 out of 12 Bomanins in total in *Drosophila melanogaster*. They have attempted to make single mutants for each of the 10 genes, but only partially succeeded in that. For some of the genes where there is no mutant fly, they have carried out experiments with ubiquitous silencing of the gene in the adult fly with Ubi-GAL4, GAL80. Conceptually, I think it is really interesting and important to try to dissect the functions of individual Bomanin genes.

The authors have obtained interesting results, but their set-up is very complex, and I think the story is very difficult to follow. This is probably because there are so many pathogens, different ways of causing infection with the pathogen (injection or natural infection), so many genes with either mutant or RNAi construct or overexpression construct, different immunological backgrounds (MyD88 deficiency, Bom_delta55C or wildtype), even different polymorphism isoforms, effect of fungal toxins... and, in some cases, not very consistent results. Biology is complex and I think it is understandable and expected that the results vary, but I would have wished for a clearer storyline and main message(s). There is a lot of data, many of which are in the supplements (extended view data), and it is really hard to dissect the main message among all the data. In addition, the discussion is really long and it is a bit hard to extract the main message(s) from that. I think maybe the main text, at least the discussion and maybe also the EV-material should be somewhat shortened to crystallize the message and main findings of the work.

Also, the nomenclature is a bit misleading at places to me. Particularly, I am not fond of the term BomTI knock-in null mutant - to me knock-in means that the gene is put back into the genome in the form of an overexpression construct. If I have understood correctly, here the authors mean that the construct containing a fluorescent marker is knocked-in to replace the gene, and therefore the gene itself is null. Although that logic is understandable, I would rather use the knock-out terminology (rather than knock-in), because the gene itself is knocked out, i.e. null. Since the authors have also used overexpression constructs in their experiments, this chosen nomenclature is even more misleading, I think.

Specific comments:

-The explanation to describe the role of Bombardier for the Bom gene expression was not easy to read. Did I understand this correctly: the authors try to explain that in the Myd88 mutant background Bbd is not expressed and therefore the Bom gene overexpression does not work in this background (Bbd is needed for stability or secretion)? Whereas in the delta55C background Bbd is expressed and the overexpression of Bom has an effect there. Please revise the text to make this point clearer (if this was the point, if I understood it correctly).

-I have to say that the logic in Table 1 is difficult to follow, at least for me. I think you should not put the explanation in the supplements (EV2) to be able to understand a Table in the main article.

-BLUD/FLUD assay: bacterial/fungal load upon death assay was not described in detail. How often were flies monitored to collect the dead flies for counting? How big a gap can there be between death and collecting the fly (I am just thinking that the bacterial counts probably rise exponentially in a fly that has been dead for a while). In methods, under the title: "Bacterial/ fungal load in single flies" in the end of the paragraph there is a mention about collecting dead flies 30 min after injection for the BLUD/FLUD assay. If the flies are collected 30 min after injection, the infection has been really short and, in my opinion, by no means reflects any differences in the immune response but rather the differences in the infection dose.

-Page 10, fourth line from the bottom up: "We have previously reported" - reference is missing

-Page 11, fifth line from the bottom up: "Of note, we formally cannot a contribution of 55C BomTs to these phenotypes" - verb missing

-P. 12, Under the title "Discussion": "Finally, we have discovered a disconnect between fungal dissemination and fungal lethality" - please rephrase, I don't know what this means

-Under "limitations of the study": "we have failed to obtain any reagent affecting..." - use some other word than reagent, it does not fit here.

-page 15, two lines above the title "The dissemination of *C. albicans* within the *Drosophila* body...": "Of note, a recent study has documented a role for BomS2..." - reference to the study missing

Minor comments:

-It would have been easier to comment specific places in the text if the lines were numbered.

-page 5, 6th row from the bottom: "we expect to have generated null mutants" - what does the sentence (the word expect) mean; is it not known/checked by e.g. qPCR if they are mutant or not? It should be relatively easy to check by qPCR if the gene is expressed in infection.

-I think in addition to Tables 1 and 2, a Table with all the fly lines in all their backgrounds used in this study should be included.
-Fig 2B: The scale is different than the rest of the survival figures - probably it would be good to change the scale also in this to 0-100%
-In Fig 2, you show the effect of natural *M. robertsii* infection, and in the text you also mention injection of spores (where there is no susceptibility of the BomT1 mutant), but this data is not shown?
Fig 3E: In what background in the Figure is the rescue with BomS2? Is it wt or MyD88 deficient, or Bom deficient? The labelling of the figure is unclear to me - in the Figure title has UAS-BomS2, does that mean that BomS2 overexpression is in all of the lines in that experiment? The rescue is impressive.
Under the title: "A role for Bomanins in preventing the dissemination of *C. albicans* throughout the fly body", fourth line from the top: DF, unpublished observations. Many journals do not allow to refer to unpublished data, not sure what the journal policy here is. It may be that this should be removed.

Referee #3:

The manuscript by Lou et al. presents comprehensive work dissecting the individual functions of the various Bomanin antimicrobial peptides in the 55C locus of *Drosophila melanogaster*. Despite the importance of antimicrobial peptides to immunity, their relative roles in defense against different pathogens or virulence factors are not fully understood. The Bomanin locus contains many peptides of unclear function, which the authors test one by one for their role against a set of bacterial and fungal pathogens. They created a panel of genetically modified flies to test either knockouts, overexpression, or RNAi against the various peptides. While not all peptides could be tested for various reasons, the panel included many of the Bomanins. Broadly speaking, the authors find that different Bomanins have a role in fighting various infections, with some distinct and some overlapping roles in these infections among the different peptides. They may also play a role in dissemination of infection throughout the host body, though it remains unclear how this may affect the host as it does not correlate with survival. Importantly, they also demonstrate that certain Bomanins are sufficient to provide protection against specific mycotoxins as well, suggesting their mechanism of protection for the hosts is fairly broad and includes both protection against spores and toxins. This work is comprehensive and it contributes to a much better understanding of the functions of these peptides. Below are comments for suggested changes or clarifications:

Broad comments:

- Please add line numbers
- Fly genetic controls
 - o In experiments where the Gal-UAS systems are used, one important genetic control appears to be missing. For example, in Figure 2C, BomT1 is overexpressed using this system, but the closest control included is the background line. So, unless genetic details on the controls were missed or not included, there are multiple genetic variables between the control and the tested line, including overexpression of the bomanin and the expression system itself. The background line alone does not account for the effect of expression of a transgene, which can impact some phenotypes in flies. While it is unlikely to change the overall results of the paper, a control where the same expression system is expressing a control gene (such as GFP) would be more ideal. It may be worth mentioning this in the caveats section.
- Citations are sometimes missing or scarce (in terms of either literature citations or citations of the figures/data being discussed). It would be beneficial to go through and make sure that any statements referencing prior literature or data within the manuscript has a citation right there. Here are a few examples:
 - o Page 4: citation for this statement needed "As reported for the entomopathogenic fungus *Beauveria bassiana*"
 - o Page 7: There is a reference to BomT1 KI mutants and injection of spores- is the citation of that data missing?
 - o Page 9: citation needed for Bombardier statement
 - o Page 15: missing citation? Sentence starting "Of note, a recent study..."
- Similarly, several methods only had citations of previous papers, but it would help to include methods in this paper as well. This is especially true for the "In vivo phagocytosis" protocol, which is not open access and I did not find a publicly available version, so the results/method could not be thoroughly reviewed due to this. Brief summaries of the major steps involved would be beneficial.
- The microscopy experiment associated with figure 4 is missing some important details.
 - o Methods for the microscopy were not in the manuscript, including the microscope used, scale bars, experimental conditions for the flies (numbers of flies examined, sex used, ages of flies, time point(s) examined after infection, etc.), and methods for measuring fluorescence. This made the results a bit unclear and difficult to review thoroughly. In particular, how were the GFP signals of the *Candida* distinguished from background fluorescence in the fly? It is difficult to distinguish background signal from the microbe
 - o Figure 4A: caption needs a bit more detail- what are the percentages? Where are the yellow arrows (an orange one is visible in one panel)?
- Data availability: suggest updating this statement- will data be available in a repository upon publication or will the authors provide data upon request?
- Were CRISPR lines checked for any off-target mutations? Although it is unlikely to be the case, were there any off-target effects (in particular, on non-target bomanins or other known AMPs) in any of the genetically-edited lines?

Minor comments:

- Page 4 and/or methods: It would help to clarify for readers which method is considered the "natural" infection method- is this the method with the needle piercing the thorax? Why would one be considered more natural than the other? It would help to address this early in the manuscript to help distinguish the methods and their relative importance for the readers
- Figure 2G: Are the myd88 flies missing from the last 2 time points due to earlier death?
- Page 8: Change "to same extent than..." to "to same extent as..."
- Figure 3E: The two shades of blue and red are difficult to tell apart- recommend shades that are more distinct
- Figure S11- please include more discussion on this- what is the suspected importance of the bristle phenotype?
- Unsure of utility of Figure S12
- Page 11: word missing in sentence starting "Of note..."
- Page 18: Mass spectrometry methods: how was single fly hemolymph collected?
- Page 19: Infection methods: what were *E. faecalis* resuspended in?
- Supplemental figures
 - o Figures S1,2,5,6,8: Clarify meaning of keys or acronyms in figures, also need statistical results on some of these figures where it is missing or unclear why it is not included
 - o Figure S2C: why were a smaller set of genes measured than other similar figures?
 - o Figure S7: are statistics missing on some of the panels?
 - o Figure S8: are statistically significant differences only shown for a subset of comparisons? Were all comparisons between all groups, or only a subgroup, made when running the statistics?
 - o Figure S8: what exactly is the y-axis ratio? Please clarify in the legend
 - o Figure S11 caption: missing D

Point-by-point reply to referees

We thank the reviewers for their careful scrutiny of our manuscript, their general appreciation of the value of our study, and their multiple suggestions to improve this manuscript. Below, you will find a point-by-point reply (in black print) to each referee comment (in blue).

Referee #1:

This manuscript represents a valuable contribution to our understanding of the role of a recently described family of peptides, the bomanins, in the antimicrobial defence of *Drosophila*. The first members of this family turned up in an early screen for immune-induced peptides in *Drosophila*, carried out by Philippe Bulet and his coworkers (Uttenweiler-Joseph et al., 1998), but a detailed study of the entire family of bomanin peptides came much later (Clemmons et al., 2015, Lindsay et al 2018). The latter authors showed that a deletion of 10 of the 12 bomanins dramatically compromised the resistance against selected bacterial as well as fungal infections. They could also restore resistance by expressing some of the Bom genes in the deficient background, and thereby getting a first indication of the relative roles of some of the peptides. Taking a slightly different approach, the authors have now addressed the same problem, using null mutants, RNAi constructs, or by overexpressing single Bom genes and testing them for resistance against a panel of microorganisms.

It should be said that this is not an easy project; to identify the individual roles for twelve different but related genes, which may have overlapping functions and which may potentially also act in concert with each other in different constellations and against different microorganisms. Therefore it is not surprising that the authors are still far from achieving that goal. Nevertheless, they have showed convincing effects for some of them, especially regarding the survival of infected animals. They have also studied the effects on the dissemination of microorganisms in the animal and the resistance against fungal toxins.

The many pleiotropic effects of the different Bom genes, affecting survival or (uncorrelated) microbe dispersion or resistance to fungal toxins is a little disturbing for small molecules like the Bomanins. The results would be easier to interpret if we had a molecular mechanism, but that would be a much bigger question and outside the scope of this communication. The authors have also recognised this problem, and they have added some speculations about it in the Discussion.

Thus, while this may not represent a breakthrough in the study of these enigmatic peptides, this manuscript is an important step forward.

However, as described below, there are still a number of questions regarding the presentation of data.

Major issues:

1. Table 1 looks rather qualitative and it is very difficult to understand. The text states that "a key to understanding Table 1 is provided in Fig. EV2", but I have not found any of the "Expanded View" (EV) figures in the package I received.

We do not understand why the referee was not able to find the EV figures that were included in the main PDF, after the main figures.

Tables 1&2 have now been included in a new Fig. 2 as they represent the core findings of our study. It is important that the reader be able to look at all of the data on a single page. As requested by Referee 2, we have also included the key to read ex-Table1 as Fig. 2B.

The data for Fig. 2A,C appear to be qualitative as they summarize a large number of survival experiments, over 800, that we analyzed. Importantly, when a difference is noted, it corresponds

to a statistically significant difference as analyzed using the Log-rank test. Thus, the data results reported in Fig. 2 are actually semi-quantitative (the extent of the difference between overexpression/"mutant" and controls is qualitative; in addition, the degree of confidence for Fig. 2C is also somewhat qualitative: see below). We have provided the corresponding survival curves for Fig. 2A (ex-Table 1) for the most relevant ones in Fig. 3C (+), Fig. 4E (+++), Figs. EV2B-F (+; ++; +++; ++,++), EV3A-D (+++; +++; ++; ++), Appendix Figs. S10A,G (ns; ++) and S11A (+) (ex-Table1); for ex-Table2: we have provided the corresponding survival curves for the most relevant ones in Fig. 3A-B,F (+++; ++), Fig4A (++) , Fig. EV2A (++) , and Appendix Fig. S10C-F (++; ns; +++; ns). Thus, the reader can judge whether our qualitative analysis is relevant. In addition, we now use a color code for Fig. 2C (ex-Table 2) to assess the degree of confidence on the phenotypes we have observed, based on whether we obtained similar results with independent lines, e.g., rather similar sensitivity in KO and KD mutants (*BomT1*), as indicated by the intensity of the green background from strong (dark) to weak (light) in Fig. 2C.

Please, see also Reply to Referee 2 on these issues.

2. Similarly, the data in Table 2 seem qualitative. What, exactly, do the sensitivity scores (++ or +++) mean, and what is the level of significance. Survival curves are more informative. Were they perhaps also provided in the missing "Expanded View" figures?

Please, see reply to point above.

3. I did not understand the comment about the MS peaks in the section about "Genetic strategies..":

"MS being semi-quantitative, it was difficult to assess the overexpression of the gene products in the context of an immune response in wild-type flies; in addition, no ectopic expression was detected in the absence of an immune challenge in flies overexpressing BomS genes"
Could that be rephrased?

We have rephrased this part in the following way: "we had to use *MyD88* flies to check that the overexpression constructs were working as expected as the signal from the overexpression strategy is masked by that from the endogenous peptides. In addition, no ectopic expression was detected in the absence of an immune challenge in wild-type flies overexpressing *BomS* genes even though they are induced at the transcriptional level (Appendix Fig. S8), suggesting that an immune challenge may be necessary for the translation or secretion of AMPs into the hemolymph".

4. Related to the point above, Figure S9A is difficult for me to interpret. I assume that the arrows point to the specific peaks of the overexpressed peptides, but in some cases their positions seem to coincide with (background?) peaks in samples where that peptide should not be expressed. Have I misunderstood?

The reviewer is quite right to point out this problem. There are indeed background peaks that appear in the 55C deletion mutant and we had to modulate our statements, as documented below at the end of the paragraph. We rigorously cannot exclude that the signal observed may be contributed by the background peaks with respects to BomS1, BomS2, and BomS5. The latter is actually masked by the Daisho short peptide (DIM4) that has been reported by the laboratory of Steven Wasserman to actually mask the BomS5 peptide (Cohen *et al.*, *Frontiers Immunol*, 2020). For the *Bomanin* genes that could not be analyzed by MALDI-TOF, we provide the RTqPCR data in addition to those already provided in our original manuscript. Our text now reads: "As expected, we then did detect the expression of several BomS peptides by MALDI-TOF MS in *Bom^{A55C}* deficiency flies (Appendix Fig. S9A). The overexpression of the other

Bom genes in the *Bom*^{Δ55C} deficiency was checked at the transcript level (Appendix Fig. S9B).” Please, note that we have modified Panel S9B to include the RTqPCR data for *BomS1*, *BomS2*, and *BomS5*.

5. In the micrographs shown in Fig. 4A, the distribution (dissemination) of *Candida* cells is difficult to distinguish from other fluorescent structures in the background.

A similar concern has also been raised by reviewer 3. We have made a novel figure, now Fig. 5, to improve it. We are now also showing blow-ups of relevant parts. Importantly, we have now added pictures for noninfected controls that we had initially omitted. No fluorescence can be observed in these controls. In the previous version, the strong background was due to the fluorescent fungus that was colonizing the fly inside the tagmata and that were outside of the plan of focus.

6. Maybe a question of editorial policy, but I think that important conclusions in the text should never rest on data that are shown only in supplementary figures or extended data files. The average impatient reader (like myself) may altogether skip the distracting and time-consuming step of switching back and forth between different documents while reading, and thereby miss the critical assessment of the strength of arguments.

We are somewhat surprised by this comment as we did place the important data in the main figures. Actually, the crux of this work is represented now by new Fig. 2, in which all of the main points of our extended analysis of survival to multiple pathogens can be visualized in a single glance. The primary data supporting this apparently qualitative analysis (see reply to point 1) are provided in the main figures 3&4 with respect to major findings resulting from the experiments monitoring the survival to infections in gain-of-function and loss-of-function *Bom* flies. Less important ones have been placed in EV figures and in Appendix supplemental figures as described above in our reply to the two first points of this reviewer. The complementary findings obtained by analyzing the dissemination of *C. albicans* and the resilience to toxins are described respectively in Figs 5&6.

Minor questions:

7. I have a problem with the commonplace generalisations about "Gram-positive" and "Gram-negative" bacteria, which are here faithfully echoed in the Introduction as well as elsewhere in this text. The Toll pathway, which is the major or only route for induction of Bomanin genes, is particularly responsive to bacterial peptidoglycans of the Lys type. Contrary to what is often assumed, such peptidoglycans are not ubiquitous among bacteria that belong to the (now rather obsolete) class of "Gram-positive" bacteria. In fact, many of the "Gram-positive" *Bacillus* and *Lactobacillus* species that are associated with insects have DAP-type peptidoglycans. This does not affect any of the conclusions in this text, but these self-perpetuating generalisations keep annoying me.

We actually fully agree with the reviewer and the senior author has always been acutely aware as documented in review articles (Ferrandon *et al.*, Nature Reviews Immunology, 2007 and Liegeois & Ferrandon, Immunogenetics, 2022). He has been in contact with a top expert in the field of peptidoglycan structure and innate immune responses, Dr. Ivo Gomperts-Boneca, who for instance told him that the DAP-type PGN of *Bacillus* is amidated in contrast to that of Gram-negative bacteria. That being said, we need to keep the Introductions simple when the study does not directly bear on the issue of sensing of infections. This is why we wrote “The major

host defense against **many** Gram-positive bacteria” to indicate that this does not fully apply to all Gram-positive bacteria. We have now rephrased it as: “The major host defense against many **but not all** Gram-positive bacteria”.

8. Please stick to standard nomenclature for mutants and other genetic constructs, with the genetic symbol (e.g. BomT1, MyD88) with a superscript for the mutant allele or other genetic variant. Here it is not always obvious if BomT1 (MyD88...) refers to a null mutant or an overexpression construct. Specifically, which MyD88 mutant was used here?

Corrected. We have now indicated the *MyD88^{c0388l}* in the Methods Section, which had been generated in the *w* [A5001] background by Exelixis (Thibaut *et al.*, Nature Genetics, 2004). To avoid any ambiguity, we now refer to “*BomOE*” (OE standing for overexpression) in the figure captions.

9. In the legend to Fig. S8, please explain "UC" (unchallenged?) , and "CS" (Canton S background?).

Done.

10. For the bacterial counts in dead flies it was not clear to me when they were sampled. I did not understand the explanation in the Material and Methods section: "Flies died in 30 minutes after injection". They must have been collected later than 30 minutes after injection, since the numbers e.g in Fig. 2E were in the range 5-25 million bacteria, much fewer than the maximum 4000 cells seen 2 hours after infection in living flies.

This huge mistake has now been corrected. The single flies were collected within 30 minutes of their deaths, which represents a considerable amount of work. Dr. David Duneau in his original eLife study (2017) had actually monitored fly death around the clock. We limited ourselves to flies dying during (long) days. This now reads: “Flies were monitored regularly and collected individually within 30 minutes after their death for bacterial load upon death (BLUD) or fungal load upon death (FLUD) analysis.”

11. Related to the point above, the base 2 logarithmic scale is not very intuitive to most of us. Base 10 logarithms are better, perhaps with the actual number specified as in Fig. 2G.

The reason for using a log₂ scale is related to microbiology: it allows to directly evaluate the number of bacterial cell divisions. The reviewer is right to point out the discrepancy in scales between different panels. For the reason stated above, we have decided to homogenize all of our microbial load panels using the log₂ scale.

12. Furthermore, the introduction of an acronym ("BLUD") for something as simple as the number of bacteria in dead flies seems unnecessary. Even more unnecessary is the additional acronym ("FLUD") for fungi. In general, acronyms should be used sparingly.

The term BLUD has been introduced by Dr. David Duneau in his seminal work (eLife, 2017) that is one of the most important conceptual advances in the field for the past years. We have introduced the term FLUD in our previous work (Xu *et al.*, EMBO Reports, 2023) to echo this important advance and believe that this will/should become a common acronym in the field, as for example is the case for DAP.

13. Apparently, "*C. glabrata*" is not closely related to *C. albicans* and is far from the genus *Candida*. Therefore, it is now renamed *Nakaseomyces glabratus* (*N. glabratus*). To avoid any misunderstanding about its relationship to *C. albicans*, please stick to the modern name. Thus,

when it is first introduced, write:

"...*Nakaseomyces glabratus* (previously called *Candida glabrata*)...", and thereafter "*N. glabratus*".

This is actually a much-debated issue and one leader in the field, Prof. David Denning, is making a rather strong case for keeping the *C. glabrata* denomination for medical mycology reasons. As we fully agree with him that “As a group fungal disease is under-recognised, under-diagnosed, under-treated, and under-funded. Splitting off additional major pathogenic species from well-established disease concepts engenders uncertainty, difficulties in messaging and hampers advocacy.” and given that “The international code of nomenclature for algae, fungi, and plants leaves open the option to retain both species and genus names for well-argued reasons. De Hoog and many knowledgeable co-authors left open the possibility to retain the name *Candida glabrata* in their sensible advice on nomenclatural stability recommendations for fungi”, we shall stick to the *C. glabrata* denomination and invite this reviewer to read the short tribune “Renaming *Candida glabrata* -A case of taxonomic purity over clinical and public health pragmatism” (Denning, D, *PLoS Biology*, 2024). We now have added in the Introduction: “(now also called *Nakaseomyces glabratus* as it is evolutionarily distant from *C. albicans*)”. Finally, we use the term candidacidal to describe the activity of BomSs against *C. glabrata*. We are not sure that the neologism “nakaseomycidal” would be easily understood, nor pronounced and widely used by other investigators.

14. In the first paragraph of the Discussion, you may have to explain the distinction between "the host defense against *M. robertsii* natural infection" and the "resistance against this infection".

We have added phrases to the first paragraph of the Discussion to allude to what resistance corresponds to. We have also added a couple of sentences in the Introduction to introduce very briefly the concepts of resistance and resilience and refer to two review articles that discuss these concepts more extensively.

Referee #2:

The authors have set out to study the individual Bomanins in the Bomanin cluster that contains 10 out of 12 Bomanins in total in *Drosophila melanogaster*. They have attempted to make single mutants for each of the 10 genes, but only partially succeeded in that. For some of the genes where there is no mutant fly, they have carried out experiments with ubiquitous silencing of the gene in the adult fly with Ubi-GAL4, GAL80. Conceptually, I think it is really interesting and important to try to dissect the functions of individual Bomanin genes.

The authors have obtained interesting results, but their set-up is very complex, and I think the story is very difficult to follow. This is probably because there are so many pathogens, different ways of causing infection with the pathogen (injection or natural infection), so many genes with either mutant or RNAi construct or overexpression construct, different immunological backgrounds (MyD88 deficiency, Bom_delta55C or wildtype), even different polymorphism isoforms, effect of fungal toxins... and, in some cases, not very consistent results. Biology is complex and I think it is understandable and expected that the results vary, but I would have wished for a clearer storyline and main message(s). There is a lot of data, many of which are in the supplements (extended view data), and it is really hard to dissect the main message among all the data. In addition, the discussion is really long and it is a bit hard to extract the main message(s) from that. I think maybe the main text, at least the discussion and maybe also the

EV-material should be somewhat shortened to crystallize the message and main findings of the work.

We agree with the referee that the biology of the 55C locus is complex and fascinating as it appears to be the major mediator of the effects of Toll pathway signaling with respect to host defense against some but not all Gram-positive infections, pathogenic yeast and filamentous fungal infections.

While the biology is complex, the conclusions we draw are “crystallized” in the Abstract that recapitulates the major findings of this study.

To address the problem of the multiple functions of 55C *Bomanin* genes, we relied on an extensive set of survival to infectious challenges in both loss-of-function and overexpression of single Bomanin genes. The most logical way is to describe the function of *Bomanin* genes in each single infection paradigm; this approach has been implemented in the Results section. We have performed some 800 survival experiments, which we cannot display in a manuscript albeit we provide the primary data in the “Source data”. The crux of this analysis is summarized in new Fig. 2. We have tried our best to make this easy to understand, even though Fig. 2A is not easy to read because it reports the data from overexpression of single Bomanin genes in three distinct genetic backgrounds, wild-type, *MyD88*, and *Bom^{A55C}*. With regards to Fig. 2C, a simple glance allows to see that we have high confidence results for the role of BomT1 against *E. faecalis* and BomS2 against injected *M. robertsii* spores since we obtained similar results for two independent mutant(s) and/or knock down RNAi line. These are shown with a dark green background. We have a lower degree of evidence with respect to BomT1 against *M. robertsii* natural infection and BomBc1 against injected *M. robertsii* spores since we respectively observed the phenotype only with the *BomT1^{Kl}* null mutant and not the knock-down line (which may be less efficient at fully silencing this gene expression) or the *BomBc1^{indel}* and again not the two knock-down lines. These are visualized with a medium-intensity green background. Finally, when we observed a phenotype with only one of the mutant/KD lines, the level of confidence is low and the corresponding cells are shaded in light green.

We have then illustrated the major results in the main figures and provided some additional survival data in the EV figures or in the Appendix figures depending on their importance. Please, see also Reply to Referee 1 for specific examples on these issues.

With respect to shortening the Discussion, we have attempted to do it but are unsatisfied with the results, as we would have to leave aside some considerations that will be of interest to the specialist. Instead, we have expanded the first paragraph of the Discussion to more thoroughly recapitulate the major findings of the study and clearly indicate at the end of this first paragraph that we propose a speculative model of Bomanin action at the end of the Discussion, which we guess will be of interest to a more general readership.

We have also removed some minor or redundant points in the Discussion.

Also, the nomenclature is a bit misleading at places to me. Particularly, I am not fond of the term BomT1 knock-in null mutant - to me knock-in means that the gene is put back into the genome in the form of an overexpression construct. If I have understood correctly, here the authors mean that the construct containing a fluorescent marker is knocked-in to replace the gene, and therefore the gene itself is null. Although that logic is understandable, I would rather use the knock-out terminology (rather than knock-in), because the gene itself is knocked out, i.e. null. Since the authors have also used overexpression constructs in their experiments, this chosen nomenclature is even more misleading, I think.

We fear that the referee may be mistaken on the definition of a knock-in mutant: it is the insertion or replacement at the original locus of the endogenous gene by a new gene so that the endogenous gene regulatory elements in the noncoding regions (introns excepted) control the

expression of the newly inserted gene like that of the replaced gene. We provide below an excerpt of a FlyBase description for an *Arc2* knock-in mutant, which is actually highly similar to the one we have generated for the *BomT1* knock-in allele in which we have replaced *BomT1* by *Gal4*: "A TI{KozakGAL4} DNA cassette has been inserted into *Arc2*, replacing the coding sequence [...]. This results in a simultaneous knock-out of *Arc2* plus a knock-in of GAL4 that is expected to be expressed under the control of the endogenous regulatory sequences of *Arc2* (predicted to gene trap all annotated transcripts of the gene). The TI{KozakGAL4} cassette was inserted via the CRISPR/Cas-9 drop-in technique". In addition, we have now modified the captions so that it is fully clear when we are overexpressing a *Bom* gene: *Bomx OE* (OE for overexpression). Please, note that overexpression is relative since the Ubiquitin promoter we have been using does not lead to as strong an expression as the endogenous promoter upon an immune challenge for most 55C *Bom* genes (Appendix Fig. S9B).

Specific comments:

-The explanation to describe the role of Bombardier for the *Bom* gene expression was not easy to read. Did I understand this correctly: the authors try to explain that in the *Myd88* mutant background *Bbd* is not expressed and therefore the *Bom* gene overexpression does not work in this background (*Bbd* is needed for stability or secretion)? Whereas in the *delta55C* background *Bbd* is expressed and the overexpression of *Boms* has an effect there. Please revise the text to make this point clearer (if this was the point, if I understood it correctly).

The referee understood the sentence perfectly right. We have nevertheless rephrased the sentences that now read: "We reasoned that the lack of detection of *BomS* peptides from overexpressed *BomS* in *MyD88* mutant flies might result from the absence of induction of other *MyD88*-dependent genes that might be required for their translation, secretion or stability in the hemolymph. Indeed, the role of Bombardier in the secretion or stability of *BomS* peptide was reported while this work was being pursued (Lin *et al.*, 2019). Importantly, *Bbd* is a Toll-regulated gene. We therefore decided to test each *Bom* gene by its overexpression for a rescuing activity of the sensitivity of *Bom*^{455C} deficiency flies to various microbial challenges, that is by testing the effects of *Bom* gene overexpression in a *Bom*^{455C} background. "

-I have to say that the logic in Table 1 is difficult to follow, at least for me. I think you should not put the explanation in the supplements (EV2) to be able to understand a Table in the main article.

We have now placed the Tables as new Fig. 2 and have included the key to Fig. 2A (ex-Table1) as panel 2B. We have also rewritten the captions to make the panel more understandable, now as legend to Fig. 2.

-BLUD/FLUD assay: bacterial/fungal load upon death assay was not described in detail. How often were flies monitored to collect the dead flies for counting? How big a gap can there be between death and collecting the fly (I am just thinking that the bacterial counts probably rise exponentially in a fly that has been dead for a while). In methods, under the title: "Bacterial/fungal load in single flies" in the end of the paragraph there is a mention about collecting dead flies 30 min after injection for the BLUD/FLUD assay. If the flies are collected 30 min after injection, the infection has been really short and, in my opinion, by no means reflects any differences in the immune response but rather the differences in the infection dose.

We apologize again for this mishap, also noted by Referee 1 (point 10). This part of the Methods section now reads: “Individual flies were collected within 30 minutes after their death for bacterial load upon death (BLUD) or fungal load upon death (FLUD) analysis.”

-Page 10, fourth line from the bottom up: "We have previously reported" - reference is missing

The reference has been inserted (Xu *et al.*, EMBO Rep., 2023).

-Page 11, fifth line from the bottom up: "Of note, we formally cannot a contribution of 55C BomTs to these phenotypes" - verb missing

Corrected. The verb “exclude” has been included.

-P. 12, Under the title "Discussion": "Finally, we have discovered a disconnect between fungal dissemination and fungal lethality" - please rephrase, I don't know what this means

Done; it now reads: “Finally, we have discovered that *C. albicans* dissemination and its lethality can be uncoupled, with again BomTs appearing to prevent to some extent the dissemination of *C. albicans*.”

-Under "limitations of the study": "we have failed to obtain any reagent affecting..." - use some other word than reagent, it does not fit here.

Done. We have replaced “reagent” by “mutant/RNAi line”.

-page 15, two lines above the title "The dissemination of *C. albicans* within the *Drosophila* body...": "Of note, a recent study has documented a role for BomS2..." - reference to the study missing

The reference has been inserted (Smith *et al.*, PLoS Pathogens, 2023).

Minor comments:

-It would have been easier to comment specific places in the text if the lines were numbered.

Done.

-page 5, 6th row from the bottom: "we expect to have generated null mutants" - what does the sentence (the word expect) mean; is it not known/checked by e.g. qPCR if they are mutant or not? It should be relatively easy to check by qPCR if the gene is expressed in infection.

In genetics, the level of expression of a gene is not sufficient to claim that we have generated a null phenotype, as we are limited by the sensitivity of the RTqPCR technique (we are most likely not sensitive to the single copy level). Thus, an apparent lack of induction does not preclude expression in a few cells such as neurons. Thus, the criterion is coming from the genomic sequence: we are certain to have a null mutant allele when the full gene sequences have been deleted, which is the case for some but not all of the mutant lines used in this study. We have actually noted that we had erroneously claimed that the *BomBC1^{indel}* mutant was a null. It is actually a mutant likely expressing a truncated protein that would belong to the BomT family of peptides since it still contains the first Bomanin domains and some additional amino-acids that form a tail. The second Bomanin domain is deleted. It is thus likely hypomorphic. We have therefore corrected the main text and added a sentence to adequately describe this mutation. RTqPCR data provide a weaker level of evidence. We have actually improved the

quality of our RTqPCR data by performing more independent experiments to strengthen the statistical analysis and added an analysis for *BomS2* null mutants that was lacking in the initial manuscript. This is also important to exclude potential off-target effects in the 55C region.

-I think in addition to Tables 1 and 2, a Table with all the fly lines in all their backgrounds used in this study should be included.

Done. This is actually a requirement of the EMBO Reports Structured Methods section.

-Fig 2B: The scale is different than the rest of the survival figures - probably it would be good to change the scale also in this to 0-100%

Done.

-In Fig 2, you show the effect of natural *M. robertsii* infection, and in the text you also mention injection of spores (where there is no susceptibility of the *BomTI* mutant), but this data is not shown?

Please, see also Reply to Referee 3. We have added the data in Appendix Figure 10B.

Fig 3E: In what background in the Figure is the rescue with *BomS2*? Is it wt or *MyD88* deficient, or *Bom* deficient? The labelling of the figure is unclear to me - in the Figure title has UAS-*BomS2*, does that mean that *BomS2* overexpression is in all of the lines in that experiment? The rescue is impressive.

The caption actually describes the genotype being tested in the experiment. We have written *w*, which corresponds to our wild-type background. There is no mention of *MyD88* nor of *Bom^{Δ55C}*. We have clarified the captions for the experimental lines by adding OE to *BomS2*.

We agree with the referee that the protection afforded to wild-type flies by *BomS2* overexpression against injected *M. robertsii* spores is especially impressive for the higher 10^7 dose.

Under the title: "A role for Bomanins in preventing the dissemination of *C. albicans* throughout the fly body", fourth line from the top: DF, unpublished observations. Many journals do not allow to refer to unpublished data, not sure what the journal policy here is. It may be that this should be removed.

This actually corresponds to observations done by the senior author some 20 years ago: these were so impressive with *C. albicans* forming hyphae at the injection site and remaining there during the whole duration of the experiment. In contrast, there was a first focus of infection at the injection site as in wild-type flies in *MyD88* flies, which was next followed by the development of additional foci of infection in other tagmata. This was the rationale for monitoring the dissemination of *C. albicans* throughout the body in *Bom^{Δ55C}* deletion and complemented mutants. We have removed this statement.

Referee #3:

The manuscript by Lou et al. presents comprehensive work dissecting the individual functions of the various Bomanin antimicrobial peptides in the 55C locus of *Drosophila melanogaster*. Despite the importance of antimicrobial peptides to immunity, their relative roles in defense against

different pathogens or virulence factors are not fully understood. The Bomanin locus contains many peptides of unclear function, which the authors test one by one for their role against a set of bacterial and fungal pathogens. They created a panel of genetically modified flies to test either knockouts, overexpression, or RNAi against the various peptides. While not all peptides could be tested for various reasons, the panel included many of the Bomanins. Broadly speaking, the authors find that different Bomanins have a role in fighting various infections, with some distinct and some overlapping roles in these infections among the different peptides. They may also play a role in dissemination of infection throughout the host body, though it remains unclear how this may affect the host as it does not correlate with survival. Importantly, they also demonstrate that certain Bomanins are sufficient to provide protection against specific mycotoxins as well, suggesting their mechanism of protection for the hosts is fairly broad and includes both protection against spores and toxins. This work is comprehensive and it contributes to a much better understanding of the functions of these peptides. Below are comments for suggested changes or clarifications:

Broad comments:

- Please add line numbers

Done.

- Fly genetic controls

o In experiments where the Gal-UAS systems are used, one important genetic control appears to be missing. For example, in Figure 2C, BomT1 is overexpressed using this system, but the closest control included is the background line. So, unless genetic details on the controls were missed or not included, there are multiple genetic variables between the control and the tested line, including overexpression of the bomanin and the expression system itself. The background line alone does not account for the effect of expression of a transgene, which can impact some phenotypes in flies. While it is unlikely to change the overall results of the paper, a control where the same expression system is expressing a control gene (such as GFP) would be more ideal. It may be worth mentioning this in the caveats section.

We do understand the referee's concerns with respect to the overexpression experiments but do not share this analysis. Indeed, Fig. 2A shows that we have even better built-in controls than overexpressing an unrelated protein, GFP, namely highly related peptides. For each line of the Fig. 2A, there is always at least one overexpression line that fails to have an effect or rescue the *Bom^{Δ55C}* immunodeficient background. For instance, for *C. glabrata*, the overexpression under our conditions (ubiquitin promoter being apparently weaker than that of endogenous *Bom* genes upon infectious challenge (Appendix Fig. 9B), *BomS1*, *BomS3*, and *BomS6* fail to rescue the *Bom^{Δ55C}* sensitivity phenotype whereas *BomS2*, *BomS4*, and *BomS5* do. The *BomS1*, *BomS3*, and *BomS6* overexpression lines are “functional” since they provide a degree of protection to *Bom^{Δ55C}* deletion flies against *A. fumigatus* (*BomS1*) or injected *M. robertsii* spores (*BomS6*) whereas *BomS4* overexpression is improving the survival to *M. robertsii* in our two infection models. An even stronger control is provided in the case of *BomBc1* since the two isoforms corresponding to distinct polymorphisms do not yield similar results in the case of *C. albicans* or *M. robertsii* spores challenges.

- Citations are sometimes missing or scarce (in terms of either literature citations or citations of the figures/data being discussed). It would be beneficial to go through and make sure that any statements referencing prior literature or data within the manuscript has a citation right there.

We apologize for these oversights, some of which had also been spotted by Referee 2.

Here are a few examples:

o Page 4: citation for this statement needed "As reported for the entomopathogenic fungus *Beauveria bassiana*"

The reference has been inserted (Hanson *et al.*, eLife, 2019).

o Page 7: There is a reference to BomT1 KI mutants and injection of spores- is the citation of that data missing?

As noted above, we have performed so many survival experiments that we cannot show each one of them in this manuscript. This is why we referred initially the reader to Table 2 for this result. We have now added the corresponding survival data to Appendix Fig. 10B.

o Page 9: citation needed for Bombardier statement

The reference has been inserted (Lin *et al.*, Front. Immunol., 2019).

o Page 15: missing citation? Sentence starting "Of note, a recent study..."

The reference has been inserted (Smith *et al.*, PLoS Pathogens, 2023).

• Similarly, several methods only had citations of previous papers, but it would help to include methods in this paper as well. This is especially true for the "In vivo phagocytosis" protocol, which is not open access and I did not find a publicly available version, so the results/method could not be thoroughly reviewed due to this. Brief summaries of the major steps involved would be beneficial.

We apologize for the omission of the *in vivo* phagocytosis protocol that we have published earlier in a technical chapter, as well as for the microscopy part with a dissection microscope at high magnification. The use of the dissection microscope made it much easier to score the large number of flies reported in Fig. 4B. The corresponding methods have now been included.

• The microscopy experiment associated with figure 4 is missing some important details.
o Methods for the microscopy were not in the manuscript, including the microscope used, scale bars, experimental conditions for the flies (numbers of flies examined, sex used, ages of flies, time point(s) examined after infection, etc.), and methods for measuring fluorescence. This made the results a bit unclear and difficult to review thoroughly. In particular, how were the GFP signals of the *Candida* distinguished from background fluorescence in the fly? It is difficult to distinguish background signal from the microbe

Please, see above for the Methods. With respect to Fig. 4, now Fig. 5, as replied to Referee 1 (comment 5), we have made a novel figure to improve it. We are now also showing blow-ups of relevant parts. Importantly, we have now added pictures for noninfected controls that we had initially omitted. No fluorescence can be observed in these controls. In the previous version, the strong background was due to the fluorescent fungus that was colonizing the fly body cavity and were lying outside of the plan of focus.

♣ Figure 4A: caption needs a bit more detail- what are the percentages? Where are the yellow arrows (an orange one is visible in one panel)?

The percentages correspond to the percentage of flies for which *C. albicans* was detected in this tagma. We are now mentioning this in the figure legend. We have removed arrows and instead provide blow-ups.

- **Data availability:** suggest updating this statement- will data be available in a repository upon publication or will the authors provide data upon request?

As per EMBO Reports requirements for a revised version, primary data are provided as “Source data”.

- **Were CRISPR lines checked for any off-target mutations? Although it is unlikely to be the case, were there any off-target effects (in particular, on non-target bomanins or other known AMPs) in any of the genetically-edited lines?**

This is an important point that we have partially already addressed in our reply to the first major comment of Referee 2. It actually also applies to RNAi lines. There are several levels of confidence. The main results of the loss-of-function concern *BomT1* and *BomS2*. For *BomT1* and *E. faecalis*, we obtained similar results using both a knock-in mutant and an independent RNAi line, which makes it highly unlikely that the observed phenotypes are due to an off-target effect, hence a high degree of confidence in the results (Fig. 2C, dark green cells). With respect to *BomS2*, similar results were obtained with two independent knock-out lines, making an off-target effect again unlikely (Fig. 2C, dark green cells). As noted earlier, we have complemented this approach by documenting by RTqPCR the level of other 55C *Bom* genes, as shown in Appendix Figures 1-6. Of note, the deletion of *BomS5* abolishes the expression of *BomT2*, but the reverse is not true. It is possible that the sequence deleted in *BomS5* is required for *BomT2* expression, independently of an off-target effect. Since we have failed to observe any increased sensitivity to the pathogens we have tested for both *BomS5* and *BomT2* null mutants, this effect does not interfere with the interpretation of the experiments. Please, note that we have also performed more RTqPCR experiments, including missing ones for *BomS2*, to increase the degree of confidence in this analysis.

Minor comments:

- **Page 4 and/or methods:** It would help to clarify for readers which method is considered the "natural" infection method- is this the method with the needle piercing the thorax? Why would one be considered more natural than the other? It would help to address this early in the manuscript to help distinguish the methods and their relative importance for the readers

The term “natural infection” has been introduced by Bruno Lemaitre in 1997 in his “furuncle (furoncle in French)” study (PNAS) in which he tested Toll pathway mutants with multiple microbial species. When testing *B. bassiana*, he shook the flies on a lawn of sporulating fungus, covering the flies with conidia. Even though it is quite common for insects to be injured in nature, with collected insects missing sometimes appendages, we believe it is appropriate to use the term “natural infection” when flies are exposed to spores in the absence of injury. The reason for this is linked to the basal biology of entomopathogenic fungi, that are actually symbionts of plants. They do protect them from herbivorous insects; *B. bassiana* protects the aerial parts of plants such as leaves or stems whereas, as their names implies, *Metarhizium* species protect the roots, by direct contact in the absence of a wound. We cannot unfortunately include this explanation in our manuscript for the sake of brevity but do hope that some readers will actually read your comments as well as our replies.

We have now included a brief explanation of the differences between the two infection models when introducing them in the paragraph entitled **Genetic strategies [...]**: “No macro-injury is

involved in *M. robertsii* natural infection as the conidia differentiate an appressorium that allow them to pierce through the cuticle and to penetrate the body cavity. The fungus exhibits distinct properties according to the infection route, both in terms of morphology (hyphal bodies in the injection vs. filaments in the natural infection) and relevant host defenses, the Toll pathway being important in both models.”

- Figure 2G: Are the myd88 flies missing from the last 2 time points due to earlier death?

This is indeed the case.

- Page 8: Change "to same extent than..." to "to same extent as..."

Corrected.

- Figure 3E: The two shades of blue and red are difficult to tell apart- recommend shades that are more distinct

Done.

- Figure S11- please include more discussion on this- what is the suspected importance of the bristle phenotype?

We have added the following text : “Thus, the constitutive expression of *BomSI* does interfere to some extent with development, and possibly elsewhere in a less visible manner. Note however that thanks to the Gal4-Gal80^{ts} system, we overexpress *Bom* genes only at the adult stage, thereby bypassing any interference with normal development.”

- Unsure of utility of Figure S12

We have deleted it.

- Page 11: word missing in sentence starting "Of note..."

Corrected.

- Page 18: Mass spectrometry methods: how was single fly hemolymph collected?

The hemolymph has been collected as in the original study from Philippe Bulet (Uttenweiler-Joseph *et al.*, PNAS, 1998) using capillarity with an empty needle.

- Page 19: Infection methods: what were *E. faecalis* resuspended in?

E. faecalis has been resuspended in PBS for injection after growth in LB. Actually, this was already indicated in the “Microbial culture and infections section” of the Methods.

• Supplemental figures

- o Figures S1,2,5,6,8: Clarify meaning of keys or acronyms in figures, also need statistical results on some of these figures where it is missing or unclear why it is not included

Done

o Figure S2C: why were a smaller set of genes measured than other similar figures?

We have extended the analysis to other 55C *Bom* genes.

o Figure S7: are statistics missing on some of the panels?

Corrected.

o Figure S8: are statistically significant differences only shown for a subset of comparisons? Were all comparisons between all groups, or only a subgroup, made when running the statistics?

The statistical comparisons that have been made are displayed for each panel of the figure.

o Figure S8: what exactly is the y-axis ratio? Please clarify in the legend

We have improved the clarity of the Y-axis in the figure legend: the data are normalized to the value measured 24h after *M. luteus* challenge in wild-type flies, taken to have an arbitrary value of 100.

o Figure S11 caption: missing D

Corrected.

Dear Dr. Ferrandon,

Thank you for the submission of your revised manuscript to our editorial offices. I have now received the reports from the three referees that I asked to re-evaluate the study, you will find below. As you will see, the referees now support the publication of your study in EMBO reports. Referees #1 and #2 have some further comments and suggestions to improve the manuscript, I ask you to address in a final revision. Please also provide a final p-b-p-response to these points and my editorial requests below.

Editorial requests:

- Please order the manuscript sections like this, using (only) these names:

Title page - Abstract - Keywords - Introduction - Results - Discussion - Methods - Data availability section - Acknowledgements - Disclosure and Competing Interests Statement - References - Figure legends - Expanded View Figure legends

- We now use CRediT to specify the contributions of each author in the journal submission system. CRediT replaces the author contribution section. Please use the free text box to provide more detailed descriptions and do NOT provide your final manuscript text file with an author contributions section. See also our guide to authors:

<https://www.embopress.org/page/journal/14693178/authorguide#authorshippinguidelines>

- Some of the scale bars in Fig. 5A are rather thin and hard to see. Please improve.

- Please check again that the number "n" for how many independent experiments were performed, their nature (biological versus technical replicates), the bars and error bars (e.g. SEM, SD) and the test used to calculate p-values is indicated in the respective figure legends. Please also check that all the p-values are explained in the legend, and that these fit to those shown in the figure. Please provide statistical testing where applicable. Please avoid the phrase 'independent experiment' but clearly state if these were biological or technical replicates. Please also indicate (e.g. with n.s.) if testing was performed, but the differences are not significant. In case n=2, please show the data as separate datapoints without error bars and statistics. See also:

<http://www.embopress.org/page/journal/14693178/authorguide#statisticalanalysis>

If n<5, please show single datapoints for diagrams. Moreover:

- Please provide the exact p values in the legends of figures 1B, C, D, E, F, G; 3A, B, C, D, E, G; 4A-E; 5B, 6A, B, C, D, F; EV2 A-F; EV3 A-D; EV4 B, C.

- Please note that the box plots need to be defined in terms of minima, maxima, centre, bounds of box and whiskers, and percentile in the legend of figure EV1.

- Please add to each legend (main, EV figures and Appendix Figures, where applicable) a 'Data Information' section (or name the provided 'notes' section like this) explaining the statistics used or providing information regarding replicates and scales. See:

- Please remove the affiliations and author names from the title page of the Appendix. It is sufficient to state 'Appendix for ...' followed by the title of the manuscript.

- Please remove the sentence 'A list of reagents is available as Supplementary Table 1'. Rather mention here the Reagents & Tools Table. Please also add callouts to the Reagents & Tools Table where appropriate.

- Thanks for providing the source data. Please upload the source data as one folder per main figure, grouping together all the files for this figure (and ZIPed together), as one folder for the EV figures and as another single folder for the Appendix figures.

In addition, I would need from you uploaded separately:

- a short, two-sentence summary of the manuscript (not more than 35 words).

- two to four short (!) bullet points highlighting the key findings of your study (two lines each).

- a schematic summary figure as separate file that provides a sketch of the major findings (not a data image) in jpeg or tiff format (with the exact width of 550 pixels and a height of not more than 400 pixels) that can be used as a visual synopsis on our website.

Please use this link to submit your revision: <https://embor.msubmit.net/cgi-bin/main.plex>

Referee #1:

This is embarrassing. I had indeed missed the fact that the "Expanded View" figures were amended at the end of the merged pdf file of the original manuscript. This was probably related to my aversion against the many different places (Supplementary, Expanded View, Supporting Data...) where important information is hidden in many journals, including EMBO Reports. I simply assumed that any Expanded View figures would be collected in a separate file. I apologise.

Anyway, I have now read the revised manuscript, with special focus on the data in Fig. 2, which are of central importance for the paper.

A key to the complicated and effectively three-dimensional table in Fig. 2A is now provided together with the figure and that is a great help. Even so, the tables and the accompanying text are still tough reading, mainly due to the complexity of the results. The data are partly contradictory and it is difficult to get a coherent picture. For instance, I find it difficult to interpret the observation that overexpression of one allelic form of BomBc1 protects the flies against injected *Metarhizium robertsii*, but not against a "natural infection" of the same pathogen, while the situation is reversed with a second allele of BomBc1, which gives no protection against injected *M. robertsii*, but partially protects against a "natural infection". Similarly, one BomBc1 null mutation makes the flies sensitive to *C. albicans* but not *M. robertsii* infection, while another one becomes sensitive to *M. robertsii* but not *C. albicans*. It is likely that these and other inconsistencies are due to the complex system and it raises the question how reproducible the results would be if they were tested under different conditions and in other genetic backgrounds. However, the authors are careful in their generalisations and they have avoided to overinterpret their results.

Other main points raised by me and the other referees have been dealt with in a satisfactory way and I still think the results are of interest for specialists in the field, even if they do not provide final answers about the role(s) of bomanins.

I still disagree on some matters of conventions and nomenclature:

1. While avoiding too frequent revisions, taxonomy should always strive to reflect phylogeny. All members of a clade (genus, family, order etc) should be more closely related to each other than to members of other clades. This is NOT a formality. It enables us to make generalisations, with some confidence, from observations made on a few species. A very good example is given by the authors in their rebuttal. The term "candidacidal" should be used for substances that kill fungi of the genus *Candida*, more or less specifically. If the substance also kills unrelated fungi such as *Nakaseomyces glabratus* it is likely to have a much wider spectrum of activity. It may even be a general fungicide.
2. Similarly, the term "Gram-positive" relates to bacteria that stain positively with the Gram staining procedure, whether they are closely related or not, and regardless of peptidoglycan structure. Generalisations about "Gram-positive" bacteria have often turned out to be misleading.
3. I agree that base 2 logarithms are informative if we want to estimate the number of cell divisions in a bacterial culture - unless there is also an on-going elimination of bacteria (lysis, phagocytosis), in which case these logarithms are not informative at all. In case we are primarily interested in how many live bacteria there are (tens, thousands, millions...), base 10 logarithms are much more intuitive and useful.
4. I strongly feel that we should avoid unnecessary acronyms as far as possible. The space saved by an acronym is usually negligible, and the loss in readability is considerable. Exceptions are in figures and table heads, where abbreviations may be useful, but they should always be explained in the caption. Genetic and chemical symbols are of course also useful.

Admittedly, this is an uphill struggle on my part. The general trend goes in the opposite direction. But, with all respect to Dr. Duneau, the terms "BLUD" and "FLUD" are particularly ugly examples. By the way, why do we need different terms when the cells are bacteria or fungi?

An example:

the sentence "We also checked the bacterial load upon death (BLUD) of single flies and found that both MyD88 and BomT1 displayed an increased BLUD in the w[A5001] background (Fig. 3E)."

could also be written

"We also checked the number of bacteria after death in single flies and found increased numbers both in MyD88C03881 and BomT1KI mutants in the w[A5001] background (Fig. 3E)."

Referee #2:

The revision has made the article substantially better and clearer. I recommend publication, but I have two more comments:

1. I still have to disagree with the authors on the point of using the terminology "knock-in" when describing the BomT1 mutant. In their response letter they dispute my criticism of the naming strategy. They have knocked in a GAL4 construct to the BomT1 site (shown in appendix Fig S1 A), but I can't find experiments where the line has been used as a driver. So to me it is misleading to describe it as a BomT1 knock-in construct. E.g. line 199: "...whether the *E. faecalis* burden was altered in BomT1 KI single flies" - this nomenclature to me suggests that BomT1 has been knocked-in, and not that this is a mutant line for BomT1. I think if the nomenclature knock-in is used, the line should rather be called BomT1-GAL4 knock-in, or something similar.

2. Another technical point related to the structure of the article: Are there two forms of "extra" material - the extended (EV) view material AND the supplemental figures? There are now 4 EV-figures in the merged manuscript file that contains the figures, and 10 supplemental figures in a separate appendix file. Seems pretty complex to me, so I wanted to double check that this is an accepted structure for a manuscript.

Referee #3:

The revisions have thoroughly addressed the questions and concerns of the reviewers. The authors have made the majority of changes that were requested. In the few cases where authors did not make suggested changes, their responses or arguments against the changes were reasonable.

Point-by-point reply to referees

Please, find below our answers to the different queries (in black print) to each referee comment (in blue).

Referee #1:

This is embarrassing. I had indeed missed the fact that the "Expanded View" figures were amended at the end of the merged pdf file of the original manuscript. This was probably related to my aversion against the many different places (Supplementary, Expanded View, Supporting Data...) where important information is hidden in many journals, including EMBO Reports. I simply assumed that any Expanded View figures would be collected in a separate file. I apologise.

Anyway, I have now read the revised manuscript, with special focus on the data in Fig. 2, which are of central importance for the paper.

A key to the complicated and effectively three-dimensional table in Fig. 2A is now provided together with the figure and that is a great help. Even so, the tables and the accompanying text are still tough reading, mainly due to the complexity of the results. The data are partly contradictory and it is difficult to get a coherent picture. For instance, I find it difficult to interpret the observation that overexpression of one allelic form of BomBc1 protects the flies against injected *Metarhizium robertsii*, but not against a "natural infection" of the same pathogen, while the situation is reversed with a second allele of BomBc1, which gives no protection against injected *M. robertsii*, but partially protects against a "natural infection". Similarly, one BomBc1 null mutation makes the flies sensitive to *C. albicans* but not *M. robertsii* infection, while another one becomes sensitive to *M. robertsii* but not *C. albicans*. It is likely that these and other inconsistencies are due to the complex system and it raises the question how reproducible the results would be if they were tested under different conditions and in other genetic backgrounds. However, the authors are careful in their generalisations and they have avoided to overinterpret their results.

I am not sure we can significantly improve the clarity of the text given the complexity of the subject, as noted by the reviewer. One has to remember that Bomanins protect the fly, to some extent, against a vast variety of microbes and associated virulence factors. For the isoform specificity, it will be highly interesting to evolutionary immunologists who will be able to determine whether there is any selection exerted on these isoforms. With respect to the differences with the infection models, we are working on a manuscript that recapitulates all the differences we have found in the two infection routes, both on the host and pathogen sides. This constitutes one more case of the importance of the infection route.

Other main points raised by me and the other referees have been dealt with in a satisfactory way and I still think the results are of interest for specialists in the field, even if they do not provide final answers about the role(s) of bomanins.

I still disagree on some matters of conventions and nomenclature:

1. While avoiding too frequent revisions, taxonomy should always strive to reflect phylogeny. All members of a clade (genus, family, order etc) should be more closely related to each other than to members of other clades. This is NOT a formality. It enables us to make generalisations, with some confidence, from observations made on a few species. A very good example is given by the authors in their rebuttal. The term "candidacidal" should be used for substances that kill fungi of the genus *Candida*, more or less specifically. If the substance also kills unrelated fungi such as *Nakaseomyces glabratus* it is likely to have a much wider spectrum of activity. It may even be a general fungicide.

We have made the changes as requested, in the text and figures. For consistency with previous studies (*e.g.*, Lindsay *et al.*, *J. Innate Immunity*, 2018), we have mentioned the former name and cited the Denning article we had mentioned in the previous round of review.

2. Similarly, the term "Gram-positive" relates to bacteria that stain positively with the Gram staining procedure, whether they are closely related or not, and regardless of peptidoglycan structure. Generalisations about "Gram-positive" bacteria have often turned out to be misleading.

As noted in our initial reply, we are fully aware of these limitations but the question is whether developing the full description with peptidoglycan structures detected by PGRP-SA/GNBP1 and PGRP-LC is going to make the introduction « heavy » and more difficult to follow. Also, this is a problem that most publications in the field have to deal with. I also note that since the reviewer comments and replies are published, the interested reader will be made aware of this issue. To us, the use of « some » Gram-positive bacteria was a « light » way to deal with this issue. The text in the first line of the Introduction now reads: "The major host defense against many but not all Gram-positive bacteria, [...]".

3. I agree that base 2 logarithms are informative if we want to estimate the number of cell divisions in a bacterial culture - unless there is also an on-going elimination of bacteria (lysis, phagocytosis), in which case these logarithms are not informative at all. In case we are primarily interested in how many live bacteria there are (tens, thousands, millions...), base 10 logarithms are much more intuitive and useful.

A base 10 logarithmic scale is now used for all microbial titers.

4. I strongly feel that we should avoid unnecessary acronyms as far as possible. The space saved by an acronym is usually negligible, and the loss in readability is considerable. Exceptions are in figures and table heads, where abbreviations may be useful, but they should always be explained in the caption. Genetic and chemical symbols are of course also useful.

Admittedly, this is an uphill struggle on my part. The general trend goes in the opposite direction. But, with all respect to Dr. Duneau, the terms "BLUD" and "FLUD" are particularly ugly examples. By the way, why do we need different terms when the cells are bacteria or fungi?

An example: the sentence "We also checked the bacterial load upon death (BLUD) of single flies and found that both MyD88 and BomT1 displayed an increased BLUD in the w[A5001] background (Fig. 3E)."

could also be written

"We also checked the number of bacteria after death in single flies and found increased numbers both in MyD88C03881 and BomT1KI mutants in the w[A5001] background (Fig. 3E)."

As requested, we have removed these acronyms and write about microbial burden at the time of death or of microbial burden within 30 minutes of death.

Referee #2:

The revision has made the article substantially better and clearer. I recommend publication, but I have two more comments:

1. I still have to disagree with the authors on the point of using the terminology "knock-in" when describing the BomT1 mutant. In their response letter they dispute my criticism of the naming strategy. They have knocked in a GAL4 construct to the BomT1 site (shown in appendix Fig S1 A), but I can't find experiments where the line has been used as a driver. So to me it is misleading to describe it as a BomT1 knock-in construct. E.g. line 199: "...whether the *E. faecalis* burden

was altered in BomT1 KI single flies" - this nomenclature to me suggests that BomT1 has been knocked-in, and not that this is a mutant line for BomT1. I think if the nomenclature knock-in is used, the line should rather be called BomT1-GAL4 knock-in, or something similar.

We have modified the notation in the main text and figure legends for both *BomT1-Gal4* (*BomT1^{KI}*) and BomT2-mCherry (*BomT2^{KI}*) KIs and have introduced then the notation used in the figures (in parenthesis in this sentence and in the text and legends.

2. Another technical point related to the structure of the article: Are there two forms of "extra" material - the extended (EV) view material AND the supplemental figures? There are now 4 EV-figures in the merged manuscript file that contains the figures, and 10 supplemental figures in a separate appendix file. Seems pretty complex to me, so I wanted to double check that this is an accepted structure for a manuscript.

This organization is the style of the journal.

Referee #3:

The revisions have thoroughly addressed the questions and concerns of the reviewers. The authors have made the majority of changes that were requested. In the few cases where authors did not make suggested changes, their responses or arguments against the changes were reasonable.

We thank the reviewer for the understanding.

Dr. Dominique Ferrandon
Université de Strasbourg/CNRS
UPR9022 Models of insect innate immunity
Strasbourg
France

Dear Dr. Ferrandon,

Thank you for the submission of your further revised manuscript to our editorial offices. I now went through this and your final p-b-p-response, and consider the remaining concerns of the referees as adequately addressed.

I am thus very pleased to accept your manuscript for publication in the next available issue of EMBO reports. Thank you for your contribution to our journal.

Yours sincerely,
